# Grain-boundary-rich polycrystalline monolayer WS$_2$ film for attomolar-level Hg$^{2+}$ sensors

Lixuan Liu[1,2,6], Kun Ye[1,6], Changqing Lin[3], Zhiyan Jia[4,5], Tianyu Xue 📧 [1✉], Anmin Nie[1✉], Yingchun Cheng 📧 [3], Jianyong Xiang[1], Congpu Mu[1], Bochong Wang[1], Fusheng Wen[1], Kun Zhai[1], Zhisheng Zhao 📧 [1], Yongji Gong 📧 [2✉], Zhongyuan Liu[1✉] & Yongjun Tian 📧 [1]

Emerging two-dimensional (2D) layered materials have been attracting great attention as sensing materials for next-generation high-performance biological and chemical sensors. The sensor performance of 2D materials is strongly dependent on the structural defects as indispensable active sites for analyte adsorption. However, controllable defect engineering in 2D materials is still challenging. In the present work, we propose exploitation of controllably grown polycrystalline films of 2D layered materials with high-density grain boundaries (GBs) for design of ultra-sensitive ion sensors, where abundant structural defects on GBs act as favorable active sites for ion adsorption. As a proof-of-concept, our fabricated surface plasmon resonance sensors with GB-rich polycrystalline monolayer WS$_2$ films have exhibited high selectivity and superior attomolar-level sensitivity in Hg$^{2+}$ detection owing to high-density GBs. This work provides a promising avenue for design of ultra-sensitive sensors based on GB-rich 2D layered materials.

[1] Center for High Pressure Science, State Key Lab of Metastable Materials Science and Technology, Yanshan University, Qinhuangdao, People's Republic of China. [2] School of Materials Science and Engineering, Beihang University, Beijing, People's Republic of China. [3] Key Laboratory of Flexible Electronics & Institute of Advanced Materials, Jiangsu National Synergetic Innovation Center for Advanced Materials, Nanjing Tech University, Nanjing, China. [4] International Collaborative Laboratory of 2D Materials for Optoelectronic Science and Technology of Ministry of Education, Engineering Technology Research Center for 2D Materials Information Functional Devices and Systems of Guangdong Province, Institute of Microscale Optoelectronics, Shenzhen University, Shenzhen, People's Republic of China. [5] International Iberian Nanotechnology Laboratory (INL), Avenida Mestre José Veiga, Braga, Portugal. [6] These authors contributed equally: Lixuan Liu, Kun Ye. ✉email: tyxue@ysu.edu.cn; anmin@ysu.edu.cn; yongjigong@buaa.edu.cn; liuzy0319@ysu.edu.cn

Atomically thin two-dimensional (2D) layered materials are emerging as structurally fascinating sensing materials for fabrication of next-generation biological and chemical sensors, such as graphene[1–3], transition-metal dichalcogenides (TMDs)[4–10], MXene[11,12], and black phosphorus[13,14]. As revealed in theoretical and experimental investigations,[2,7,8,15–17] the structural defects in 2D layered materials, such as vacancy, antisite, substitution, edge, grain boundary (GB), etc., are able to significantly alter its physico-chemical properties, including energy landscape, chemical reactivity and selectivity. As energetically activated, chemically selective sites, the structural defects are able to ultimately govern the 2D material based sensor selectivity and sensitivity. Therefore, the controllable high-throughput production of structural defects is vital in exploitation of various 2D layered materials as ultra-sensitive sensing materials. Though the structural defects in 2D layered materials can be formed during the fabrication process or via defect engineering post treatments,[18–23] the controllable high-throughput introduction of defects is still difficult and remains as a big challenge.

As reported in the previous investigations[24–29], grain boundaries (GBs) as the typical structural defects are able to induce intrinsic activation of the 2D basal plane, and thus their presence leads to the application potential of 2D materials in many fields including solar cells[30], electrocatalysis[28,31], sensors[16,32], etc. Importantly, GBs of high density can be introduced during growth of polycrystalline monolayer (1L) TMDs films in centimeter or wafer scale.[33–37] As the fascinating members among the large family of TMDs, monolayer (1L) W(Mo)S$_2$ have been intensively investigated in recent years because of their excellent properties, such as direct band gap, strong light-matter interaction, mechanical flexibility, high environmental stability, etc. Recently, we have reported the controllable CVD growth of GB-rich polycrystalline 1L W(Mo)S$_2$ films.[34,35] The sharply increased density of GBs has been achieved via growth of the nanoscale 1 L W(Mo)S$_2$ grains in a narrow size distribution around an average size of ~40 nm. On GBs in 1L W(Mo)S$_2$ films, there exist rich active S sites. The high-density GBs thus provide huge amounts of desirable active sites for preferential or selective adsorption of mercury ions (Hg$^{2+}$, one notorious heavy metal pollutant in water). In addition, W(Mo)S$_2$ shows excellent stability in air or solution, which is beneficial for application of the GB-based sensor in complex solutions and long shelf time without strict requirements for storage conditions. Thereby, the GB-rich polycrystalline 1L W(Mo)S$_2$ film as sensor material is expected to exhibit superior sensing response in Hg$^{2+}$ detection.

In this paper, we demonstrate the significant role of high-density GBs in design of high-performance ion sensors based on 2D materials. As a proof-of-concept, we fabricated the GB-based surface plasmon resonance (SPR) Hg$^{2+}$ sensors by using GB-rich polycrystalline 1L W(Mo)S$_2$ films as sensing materials. Our investigations have indicated the enhanced Hg$^{2+}$-detection sensitivity of GB-based SPR sensor to attomolar level and the detection limit of 1 aM owing to the presence of high-density GBs. Theoretically and experimentally, the selective adsorption of Hg$^{2+}$ on GBs has been revealed to occur via the S–Hg bond formation. Most directly, our studies provide a representative demonstration of the potential application of GB-rich polycrystalline one or few-layer film of any layered metal sulphides as sensing materials in highly sensitive Hg$^{2+}$ detection. Furthermore, exploitation of GB-rich polycrystalline one or few-layer film of any layered materials can be inspired to act as potential sensing materials in the sensitive detection of broader types of analytes, such as biomolecules, metal ions, gases, etc.

## Results
### Growth and characterization of polycrystalline WS$_2$ film. In the polycrystalline films of 2D layered materials, the density of GBs,

i.e., the total GB length per unit surface area, is controlled by the lateral grain sizes and their distribution. The sharply increased density of GBs occurs only for the growth of grains in the nanoscale sizes (Supplementary Figs. 1–4 and details in Supplementary discussion), implying the immensely increased structural defects or active sites per unit surface area. Thereby, the growth of nanoscale grains is desirable for the potential application of GB-rich polycrystalline 2D layered films as ultra-sensitive sensing materials.

In our CVD-grown centimeter-scale polycrystalline 1L WS$_2$ films, the grains were controllably grown in a narrow size distribution and the average size is just ~40 nm, as confirmed by measurements of optical microscopy (OM), fluorescence (FL), atomic force microscopy (AFM), Raman and photoluminescence (PL) (Supplementary Figs. 5 and 6, and details about CVD growth and characterizations in "Methods"). Figure 1a and b show the low and high magnification AFM images of the as-grown polycrystalline 1L WS$_2$ film on Si substrate after 1-week exposure to air for sufficient adsorption on GBs. The GB profiles can be clearly visualized between irregular-shaped WS$_2$ grains with the lateral dimension of less than 100 nm in average. The observable GBs in AFM images is due to the preferential adsorption of contaminants on GBs under exposure to air.[33] The nanoscale grain sizes are the direct evidence of high-surface-density GBs in CVD-grown polycrystalline 1L WS$_2$ film. The atomic structure of GBs was investigated in detail by using scanning transmission electron microscopy (STEM). The low magnification high angle annular dark field STEM (HAADF-STEM) image of Fig. 1c clearly shows the GBs and the nanoscale grains with lateral sizes of less than 100 nm, confirming the AFM observations. In the corresponding selected area electron diffraction (SAED) of Fig. 1d, the observed diffraction rings instead of individual spots suggest that the crystal grains in polycrystalline 1L WS$_2$ film were grown in nanoscale lateral sizes and without a preferential orientation. Statistical analysis of the grains indicates that the lateral size dominantly lies in the range from 20 to 55 nm, as shown by the inset of Fig. 1c, demonstrating the high grain density ($10^{10}$–$10^{11}$ cm$^{-2}$) and high GB surface density. Atomic structure of the GBs on the as-synthesized WS$_2$ film is distinguished using atomic-resolution HAADF-STEM imaging. After systematic examination of multiple boundary locations (Supplementary Fig. 7), we confirmed that the WS$_2$ grains in the as-synthesized film are indeed stitched together via GBs. It is established in previous research that the atomic make-up of GBs is diverse. In our sample, the GBs primarily consist of 5- and 7-fold (5|7) rings with a sulfur-rich chemical composition, as illustrated in Fig. 1e.

### Hg$^{2+}$ sensing of GB-rich 1L WS$_2$ film. The GB-based SPR sensors were fabricated with GB-rich polycrystalline 1L WS$_2$ film as sensor material to demonstrate the significant role of GBs in the high sensor performance in Hg$^{2+}$ detection (fabrication details in "Methods"). The basic setup of our sensor device and the SPR imaging system are schematically illustrated in Supplementary Fig. 8. The detection principle of SPR sensors lies in the ultra-sensitivity of SPR signals (i.e., resonance angle $\theta$ in the present work) to minute changes in refractive index of the sensing surface. During detection, Hg$^{2+}$ adsorption on the GBs leads to change in the refractive index of 1L WS$_2$ film as sensor material, and the induced shift of SPR curve provides the quantitative information of captured Hg$^{2+}$ ions on 1L WS$_2$ film. By using Snell's Law and the N-layer transfer matrix method, we performed the computational simulation to profile the vertical electric field distribution of Au film before and after the transferring of a 1L WS$_2$ film (Supplementary Fig. 9). Compared with bare Au, the electric field is significantly enhanced upon the

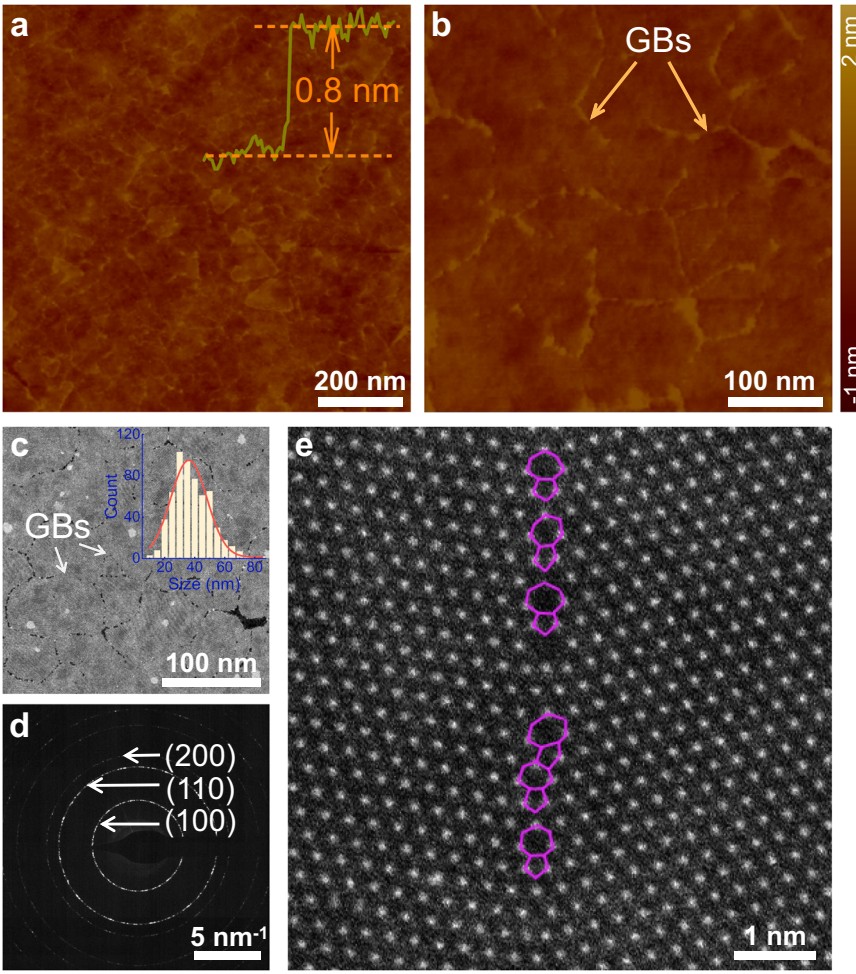

**Fig. 1 Characterization of grain boundaries (GBs) in CVD-grown polycrystalline 1L WS$_2$ film. a, b** AFM images of low and high magnifications. The height profile was taken from a dash line as shown in Supplementary Fig. 5c. **c, d** Low-magnification HAADF-STEM image and SAED pattern of as-grown polycrystalline 1L WS$_2$ film. Inset in **c** is the distribution of grain sizes. **e** Atomic scale HAADF-STEM image of a typical GB in the polycrystalline 1L WS$_2$ film, primarily composed of 5|7 rings (purple lines).

incorporation of GB-rich 1L WS$_2$, and it is further increased to a maximum at the interface between 1L GB-rich WS$_2$ film and sensing medium. The 1L GB-rich WS$_2$ integration induced electric field enhancement between Au film and sensing medium signifies that the proposed SPR sensor is sensitive to slight changes in sensing medium, thus strongly suggesting the suitability of 1L WS$_2$ film for SPR sensor. Generally, SPR signal optimization can be induced via incorporation of many other 2D layered materials onto Au film,[38,39] which is favorable for detection sensitivity. However, this effect itself is insufficient for high sensor performance, and the fundamentally indispensable key factor is the high analyte adsorption ability of sensor material.

To demonstrate the significant role of GBs in physi and/or chemi-adsorption of Hg$^{2+}$, the CVD-grown 1L WS$_2$ single crystals (Supplementary Fig. 10) based SPR sensor was also evaluated in Hg$^{2+}$ detection. Compared with the rich structural defects along GBs in polycrystalline 1L WS$_2$ film, the 1L WS$_2$ single crystal has much fewer structural defects on the surface, but possesses edge defects. As revealed in the previous studies,[7] for the biochemical sensors based on 2D layered crystals, preferential adsorption occurs along crystal edges. Thus, the Hg$^{2+}$ sensing abilities of SPR sensors based on GB-rich 1L WS$_2$ film and 1L WS$_2$ single crystal were investigated in detail for comparison. Figure 2a shows the angle-resolved SPR spectra of GB-rich 1L WS$_2$ film and 1L WS$_2$ single crystal in ultrapure water and at increasing concentrations of Hg$^{2+}$ solution ($10^{-18}$–$10^{-11}$ M).

In contrast to those of the SPR sensor based on 1L WS$_2$ single crystal, the SPR spectra of GB-based sensor display much more prominent right-shift with increasing Hg$^{2+}$ concentration, indicating adsorption of more Hg$^{2+}$ ions. Figure 2b shows the determined resonance angle shift ($\Delta\theta$) as a function of Hg$^{2+}$ concentration for the SPR sensors based on GB-rich 1L WS$_2$ film and 1L WS$_2$ single crystal for comparison. Distinctly, at the same Hg$^{2+}$ concentration, the GB-based sensor exhibits a much larger angle shift ($\Delta\theta$) than that of the 1L WS$_2$ single crystal. The degree of angle shift is proportional to the amount of adsorbed Hg$^{2+}$. Even at attomolar-level concentration (13 milli-degree for $10^{-18}$ M), the adsorption of Hg$^{2+}$ by the GB-rich 1L WS$_2$ film brings about discernible change in SPR resonance angle. Notably, compared with the 1L WS$_2$ single crystal based sensor, the GB-based sensor displays a much wider detectable range of Hg$^{2+}$ concentration. For the GB-based SPR sensor, the most sensitive response occurs in the concentration range from $10^{-17}$ to $10^{-13}$ M ($\Delta\theta = 26.7 \cdot \log[\text{Hg}^{2+}] + 477.5$, $R^2 = 0.9801$). Below $10^{-17}$ M, the sensor response becomes weaker and exact Hg$^{2+}$ quantification is not easy, but, semi-quantitative analysis regarding the order of detected Hg$^{2+}$ concentration is still achievable from the SPR resonance angle shift, as it is still discernible even at the attomolar-level concentration (13 milli-degree for $10^{-18}$ M). Above $10^{-13}$ M, the response tends to be saturated. The drastically enhanced performance of GB-based sensor reveals the significant role of high-surface-density GBs in sensor performance.

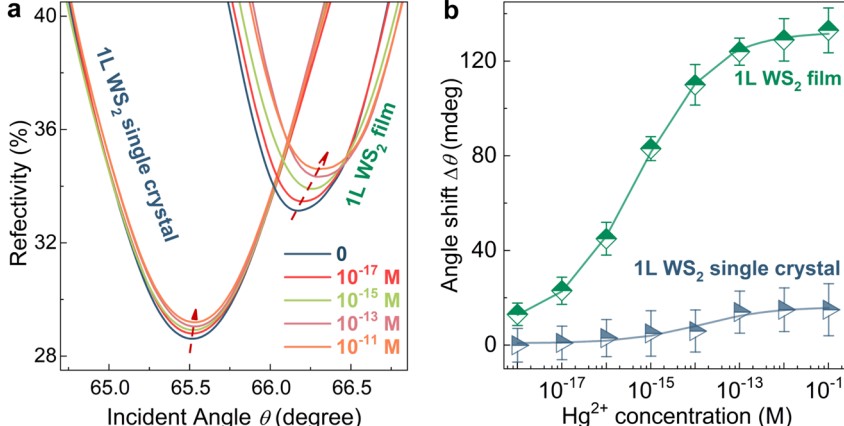

**Fig. 2 Hg²⁺ detection of SPR sensor based on GB-rich 1L WS₂ film and 1L WS₂ single crystal. a** SPR spectra at Hg²⁺ concentrations ranging from $10^{-18}$ M to $10^{-11}$ M. **b** Corresponding extracted resonance angle shifts of $\Delta\theta$. For the prepared $10^{-18}$ M Hg²⁺ concentration, the x-axis error bar was estimated to be $\pm 0.1 \times 10^{-18}$ M ("Methods"). At each Hg²⁺ concentration, the collection of data was performed for 5 times, from which the average value of $\Delta\theta$ and y-error bar were determined.

**Selective adsorption of Hg²⁺ on GBs.** The structural defects in 2D layered materials, such as vacancy, antisite, substitution, edge, and GBs, are widely accepted as energetically active positions for adsorption or binding sites.[7,16,17,24–31,40–47] For fundamental understanding on the significant role of GBs in Hg²⁺ adsorption, we performed DFT calculations by considering the chemisorption of Hg²⁺ ions on GBs and GB-free area for comparison, as shown in Fig. 3a, b and Supplementary Fig. 11 (calculation details in "Methods"). The adsorption energies $E_{ads}$ of Hg²⁺ ions on different positions around GBs are listed between Fig. 3a and b. It is evident to find that Hg²⁺ prefers to adsorb on the pentagon hollow sites along GBs. Moreover, the adsorption energies of Hg²⁺ around the GBs are also lower than that on pristine 1L WS₂ crystal (Supplementary Fig. 11). This indicates the GBs behave like the Hg²⁺ traps. From the charge density difference plot shown in Fig. 3b, we can find the apparent charge transfer between Hg²⁺ and GBs. There are 0.1 negative charge transfer from Hg²⁺ to neighboring S atoms, indicating the formation of covalent bonds between Hg and S atoms. The formation of S–Hg bond is substantiated by X-ray photoelectron spectroscopy (XPS) measurements of the as-synthesized GB-rich 1L WS₂ film before and after Hg²⁺ detection. Figure 3c–h show the XPS spectra of W 4f, S 2p and Hg 4f core levels before and after Hg²⁺ detection. For the as-synthesized GB-rich 1L WS₂ film, the two deconvoluted W 4f peaks at 32.63 and 34.79 eV (Fig. 3c) and the two deconvoluted S 2p peaks at 162.55 and 163.71 eV (Fig. 3d) are ascribed to WS₂,[48] while no trace of Hg is observed (Fig. 3e). For the GB-rich 1L WS₂ film after Hg²⁺ detection in $10^{-9}$ M Hg²⁺ solution and then rinsed with ultrapure water for three times, obvious sign of S–Hg bonds is observable in the XPS spectrum of Hg 4f core level (Fig. 3h), and the main deconvoluted peaks of W 4f and S 2p are slightly shifted toward higher binding energies in contrast to those of the as-synthesized GB-rich 1 L WS₂ film. The two deconvoluted W 4f peaks at 33.05 and 35.22 eV can be still related to WS₂ (Fig. 3f). On the XPS spectra of S 2p core level, however, in addition to the two strong deconvoluted peaks at 162.55 and 163.71 eV from WS₂, two new weak deconvoluted peaks at 161.67 and 163.11 eV are recognized to be produced by the formation of S–Hg bonds during Hg²⁺ detection (Fig. 3g). The formation of S–Hg bonds after Hg²⁺ detection, which can induce changes in chemical environment for W and S atoms in the WS₂ film, can also be the underlying origin for the observed slight shift of the XPS W 4f and S 2p peaks from WS₂ toward higher binding energies. The combined theoretical calculations and XPS measurements provide concrete evidence for the preferential adsorption of Hg²⁺ on the GBs via the formation of S–Hg bonds. Therefore, the

rich chemically active sites on GBs in polycrystalline 1L WS₂ film serve as efficient probes for ultra-sensitive detection of Hg²⁺ ions.

**Sensor performance.** As one important performance criterion, selectivity of the GB-rich WS₂ SPR sensor for Hg²⁺ detection was carefully assessed. In view of the revealed Hg²⁺ adsorption via S–Hg bond formation, common interfering ions with possible affinity for sulphur were evaluated, including Pb²⁺, Ag⁺, Zn²⁺, Fe³⁺, Ca²⁺, and Mg²⁺. With the $10^{-15}$ M Hg²⁺ solution (S0) being used as control, the interference studies were performed in a series of S1–S7 solutions with $10^{-15}$ M Hg²⁺ ions and the interfering ions of different concentrations. In solutions of S1(S2), the Pb²⁺(Ag⁺) concentrations are varied from $10^{-15}$ M to $10^{-12}$ M, and in solutions of S3–S6, the concentrations of Zn²⁺, Fe³⁺, Ca²⁺, or Mg²⁺ are varied from $10^{-8}$ M to $10^{-5}$ M. Solution S7 contains $10^{-15}$ M Hg²⁺ and the mixed interfering ions of Pb²⁺, Ag⁺ at $10^{-12}$ M and Zn²⁺, Fe³⁺, Ca²⁺, Mg²⁺ at $10^{-5}$ M. Figure 4 shows the determined values of $\Delta\theta$ from sensor responses to the series of S0–S7 solutions.

Compared with that of S0 ($10^{-15}$ M Hg²⁺) as reference, the sensor responses to the S1(S2) solutions exhibit just a slight increase in resonance angle shift $\Delta\theta$ when the Pb²⁺(Ag⁺) concentration is increased from $10^{-15}$ M to $10^{-12}$ M, implying the negligible dependence of $\Delta\theta$ on Pb²⁺(Ag⁺) concentration and thus the negligible interference of Pb²⁺(Ag⁺) ions to Hg²⁺ detection of the GB-based sensors. This is further corroborated with XPS analysis of sensor material after detection in mixed Hg²⁺ ($10^{-15}$ M) and Pb²⁺ ($10^{-12}$ M) solutions. The XPS spectra exhibit the appearance of a distinct Hg 4f core level peak around the binding energy of 102 eV but no observable sign of Pb 4f core level peak around the binding energies of ~138 and ~143 eV (Supplementary Fig. 12). Thereby, the interference to Hg²⁺ detection from the coexisting Pb²⁺(Ag⁺) ions could be neglected even at a 1000× higher concentration of interfering ions. Additionally, compared with S0 ($10^{-15}$ M Hg²⁺) as reference, the S3–S6 solutions lead to no significant variations of $\Delta\theta$ when the concentrations of Zn²⁺, Fe³⁺, Ca²⁺, or Mg²⁺ ions are increased from $10^{-8}$ M to $10^{-5}$ M. Even in the presence of mixed interfering ions, the S7 solution does not produce an obvious change in the sensor response in comparison to the reference (S0), and just a negligible increase in $\Delta\theta$ is observable. These detailed results indicate the satisfactory selectivity for Hg²⁺ detection of the GB-based sensor.

Overall, the above interference studies reveal that the GB-rich polycrystalline 1L WS₂ film is highly selective towards absorption

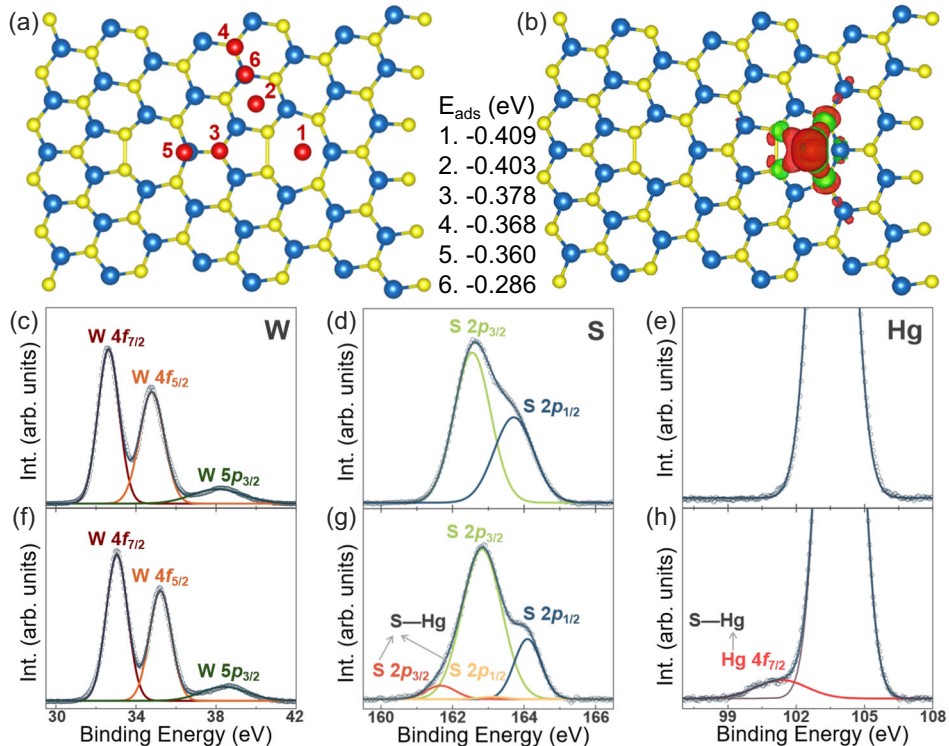

**Fig. 3 DFT calculations and XPS characterization. a** DFT calculations regarding different adsorption positions of $Hg^{2+}$ on grain boundary labeled out with number 1–6. **b** Charge density difference between $Hg^{2+}$ and $WS_2$ with grain boundaries. The calculated adsorption energies of $E_{ads}$ at the 6 different positions are listed between (**a**, **b**). Yellow, blue, and red spheres represent S, W, and Hg atoms, respectively. There is apparent charge transfer between $Hg^{2+}$ and nearby S atoms. The isosurface value is 0.0002 $e/Bohr^3$. **c–e** XPS spectra of W $4f$, S $2p$ and Hg $4f$ core levels for the as-synthesized 1L $WS_2$ film. **f–h** And corresponding XPS spectra of the 1L $WS_2$ film after $Hg^{2+}$ detection. Notably, the XPS peak at 104.54 eV in (**e**, **h**) is ascribed to Si from wafer.

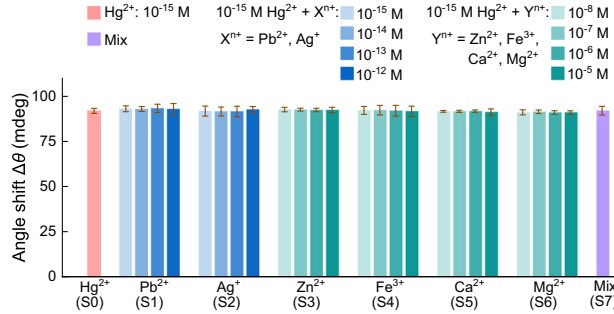

**Fig. 4 Interference study of $Hg^{2+}$ detection.** GB-based SPR sensor response ($\Delta\theta$) to $Hg^{2+}$ at a fixed concentration ($10^{-15}$ M) with increasing concentration of interfering ions ($Pb^{2+}$, $Ag^+$, $Zn^{2+}$, $Fe^{3+}$, $Ca^{2+}$, and $Mg^{2+}$). S0 is the $10^{-15}$ M $Hg^{2+}$ solution as reference. S1(S2) are the solutions of $10^{-15}$ M $Hg^{2+}$ and $Pb^{2+}(Ag^+)$, the $Pb^{2+}(Ag^+)$ concentration is varied from $10^{-15}$ M to $10^{-12}$ M. S3–S6 are the solutions of $10^{-15}$ M $Hg^{2+}$ and one of the naturally abundant interfering $Zn^{2+}$, $Fe^{3+}$, $Ca^{2+}$, and $Mg^{2+}$ ions, the $Zn^{2+}$, $Fe^{3+}$, $Ca^{2+}$, $Mg^{2+}$ concentration is varied from $10^{-8}$ M to $10^{-5}$ M. S7 is the solution of $10^{-15}$ M $Hg^{2+}$ ions and mixed interfering ions of $Pb^{2+}$, $Ag^+$ at $10^{-12}$ M and $Zn^{2+}$, $Fe^{3+}$, $Ca^{2+}$, $Mg^{2+}$ at $10^{-5}$ M. All error bars is the standard deviation of SPR angle shift from five repeated measurements.

of $Hg^{2+}$ over possible interfering ions. In consideration of the $Hg^{2+}$ absorption via S–Hg bond formation on GBs, the observed selectivity can be well understood with the hard-soft-acid-base (HSAB) theory.[49] In metal-sulphur bonding, a typical strong lewis acid and base soft-soft interaction, Sulphur is the soft base with binding preference for soft acids. This explains the high affinity of GB-rich polycrystalline 1L $WS_2$ film towards $Hg^{2+}$, as

it is a typical soft acid; and the negligible sensing interference from borderline and hard acids like $Pb^{2+}$, $Ag^+$, $Zn^{2+}$, $Fe^{3+}$, $Mg^{2+}$, $Ca^{2+}$, even at the higher concentration. In short, the specific soft-soft interaction explains the experimentally observed high selectivity of GB-rich polycrystalline 1L $WS_2$ towards $Hg^{2+}$.

Investigations were also performed on the accuracy, repeatability, stability and reusability of the GB-based SPR sensor (Supplementary Figs. 13–15). In addition to the high selectivity, the GB-based SPR sensor also exhibits excellent stability (>6 month shelf life of GB-rich $WS_2$ film and no decay during measurement), good repeatability and a better accuracy of ~0.4 than that of graphene-based SPR sensor (~0.2, 680 nm laser light). As for the reusability, the Au film substrate can be recycled, but the GB-rich 1L $WS_2$ film as sensing material cannot be reused due to covalent adsorption of $Hg^{2+}$. Additionally, no intentional acidification was performed in all the prepared $Hg^{2+}$ solutions. Their pH lies at ~6.1, which does not impact sensor response (Supplementary Fig. 16).

To further validate the superior sensing capacity of GB-rich $WS_2$ film, we listed the limit of detection (LOD) of $Hg^{2+}$ for GB-based sensor and those of previously reported $Hg^{2+}$ sensing materials (Supplementary refs. 11–74) in Fig. 5 for comparison. The LOD of GB-rich 1L $WS_2$ film in $Hg^{2+}$ detection can reach to 1 aM (~ 600 ions $cm^{-3}$), according to the IUPAC guideline of 3:1 signal to noise ratio, which clearly outperforms the previously reported $Hg^{2+}$ sensors based on conventional sensor materials. Additionally, to evaluate the practical applicability of GB-based sensor, examination was performed on the sensor responses to a series of standard $Hg^{2+}$ solutions ($10^{-18}$–$10^{-11}$ M), which were prepared by using tap water filtered with filter paper (Supplementary Fig. 17 and detailed discussion in Supplementary information). The three tested GB-sensors give robust and quite

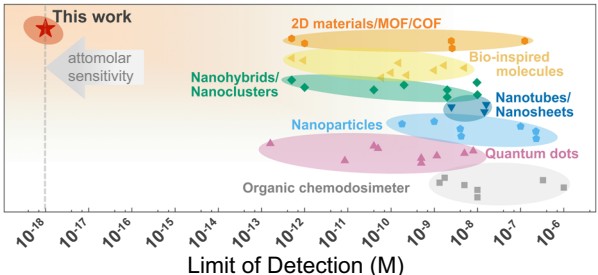

**Fig. 5 Limit of detection comparison.** Overview of the $Hg^{2+}$ detection limit by using different sensor materials (Supplementary refs. 11–74).

consistent responses to the prepared solutions of $Hg^{2+}$ ions, and a clear positive correlation between $Hg^{2+}$ concentration in tap water and sensor response can be achieved. The corresponding average value of $\Delta\theta$ can be expressed by an empirical formula of $\Delta\theta = 19.2*\log[Hg^{2+}] + 356.7$ ($R^2 = 0.9820$) within the $Hg^{2+}$ concentration range of $10^{-17}$ M to $10^{-13}$ M. Compared with that in ultrapure water (26.7), though the slope of 19.2 in tap water becomes smaller, it still keeps the same order, and more importantly, an obvious value of $\Delta\theta$ is still observable even at 1 aM level of $Hg^{2+}$ ions, implying the good practical applicability of GB-based sensors. Notably, tap water involves not only the possible metal ions but also the other more complex contaminants such as organic impurities, germs, viruses, etc. Though the simple filtration with filter paper can remove most of the insoluble impurities in tap water, the observable deterioration of sensor response ($\Delta\theta$) to $Hg^{2+}$ ions can still be produced by the interferences from some other more complex contaminants. The sensor response in tap water would be further optimized if the more complex contaminants and their possible interferences were able to be differentiated.

The superior performance of GB-rich 1L $WS_2$ based SPR sensor in $Hg^{2+}$ detection relies on the presence of sufficient active S sites along the rich GBs for the preferential and efficient adsorption of $Hg^{2+}$ via S–Hg bond formation. In view of the theoretically and experimentally revealed adsorption mechanism, similar to GB-rich polycrystalline 1L $WS_2$ film, GB-rich polycrystalline films of any other layered metal sulphides would also be expected to serve as sensor materials for ultra-sensitive detection of $Hg^{2+}$ ions. As an additional demonstration, the CVD-grown GB-rich polycrystalline 1L $MoS_2$ film[35] was used to fabricate the SPR sensor for detection of $Hg^{2+}$ ions. As shown in the SPR spectra and the determined resonance angle shifts of $\Delta\theta$ (Supplementary Fig. 18), the superior sensor sensitivity to attomolar-level quantification (LOD of 1 aM) is also observed for the GB-rich 1L $MoS_2$ film, comparable to those of the GB-rich 1L $WS_2$ film. Thereby, the concept of GB-rich 2D layered materials for high-performance sensors should be universal and comprehensive.

It is of significance to find that the GB-based SPR sensor is able to reach such a low detection limit from a fundamental research perspective, because it demonstrates the great potential of GBs of 2D materials in fabrication of sensitive, low-limit ion sensors. The rich GBs provide abundant active sites for analyte adsorption and the ultrathin thickness of 2D materials makes them very sensitive to the adsorbed species, which is complementary to traditional sensor materials.

## Discussion

In summary, we report the fabrication of GB-based SPR sensors with CVD-grown GB-rich polycrystalline 1L $WS_2$ films and their superior sensing performance in $Hg^{2+}$ detection. High surface

density GBs supply abundant structural defects or active S sites for efficient $Hg^{2+}$ absorption via S–Hg bond formation as revealed by the combined DFT calculations and XPS measurements, thereby promoting the attomolar $Hg^{2+}$ detection of the GB-based SPR sensors ($Hg^{2+}$ LOD of 1 aM). As crucial sensor performance criteria, the GB-based SPR sensors exhibit not only high selectivity but also excellent accuracy, stability, and repeatability. Our studies on the GB-rich 1L $WS_2$ film based SPR sensors will directly stimulate the exploitation of GB-rich polycrystalline mono- and few-layer films of any other layered metal sulphides as sensing materials for fabrication of ultrasensitive $Hg^{2+}$ sensors, where similar high performances are expected. Furthermore, our work demonstrates the significance in further exploiting the potential of GB-rich 2D materials as sensor materials for detection of broader ranges of analytes.

## Methods

**CVD growth of polycrystalline 1L $WS_2$ film.** The centimeter-scale polycrystalline 1L $WS_2$ and $MoS_2$ films were grown on $SiO_2$ (300 nm)/Si substrates in a homemade two-zone furnace with a 60 mm diameter quartz tube as reported in our previous works[34,35]. Two 1.3 mm diameter inner quartz tubes were used as carrier gas (Ar) pathways for delivery of the S and W sources. For $WS_2$, the two temperature zones were heated to 930 °C and 800 °C in 40 min. The $H_2S$ gas as S source (50 sccm) and $WO_3$ (0.5 g, Alfa Aesar, purity 99.99%) vapors as W source were carried by Ar gas in two individual pathways (20 and 50 sccm, respectively) to the substrate, and the pressure inside the tube was maintained at about 0.1 kPa. The reaction takes 30 min, after which the furnace was naturally cooled for 15 min, and then opened for quick cool down. The tail gas was filtered through NaOH (sat. aq.). $MoS_2$ synthesis follows a similar procedure, but the two temperature zones were set at 450 °C and 925 °C, respectively.

**Material characterization.** OM and FL images were acquired with an optical microscope (DM4000 M, Leica). Raman and PL spectra, line scans and mapping images were obtained with a Horiba Jobin Yvon LabRAM HR-Evolution Raman microscope under 532 nm laser irradiation. AFM characterizations were performed on Multimode 8 (Bruker). TEM, HAADF, and SAED measurements were performed using a transmission electron microscope (Talos F200X, Thermo Fisher) operated at 200 kV. XPS spectra were acquired using an Axis Ultra spectrometer (ESCALAB 250Xi, Thermo Fisher).

**Fabrication of SPR sensor device.** SPR sensor devices are constructed of the as-synthesized TMDs films transferred onto a 47 nm Au film-coated cover glass substrate (MATSUNAMI GLASS, Japan; Supplementary Fig. 5a). Before deposition of Au film, 2 nm Cr film was first deposited for the enhanced adhesion of Au film to cover glass substrate. The transfer of as-grown TMDs films were performed via a standard PMMA-based wet transfer method. PMMA was spin-coated onto the as-synthesized film, baked at 150 °C for 180 s, and immersed in 100 °C $H_2O$ for 10 min to detach the PMMA-coated film from the substrate. The Au-coated cover glass substrates were used to pick up the PMMA-coated film. Finally, by dissolving the PMMA in acetone, followed by 3× ethanol rinse and 3× DI water rinse, the as-synthesized films were successfully transferred.

**Experimental setup and sensor characterization.** The adopted SPR imaging system was based on an inverted microscope (Nikon TI2-E, Japan) and a 100× (NA 1.49) oil immersion objective. A 680 nm 10 mW He-Ne laser was used as the light source to excite surface plasmon on the gold surface. The CMOS camera (Zyla 4.2 PLUS, Andor, Belfast, UK) was used to record the SPR image at different incident angles. A stepping motor was incorporated on the optical fiber to translate the incident angle of light beam. The incident angle was synchronized with the corresponding SPR image. The intensity measurement was performed using Image J software.

A silicon insert (flexiPERM®8-well reusable silicon insert, Sarstedt, Germany) is placed on the sensor device to hold analyte solutions (inner dimensions: 11 mm × 8 mm × 9 mm). Angle-resolved SPR spectra was acquired at room temperature (around 23 °C) in ultrapure water and then at increasing concentrations of $Hg^{2+}$ aqueous solutions ($10^{-18}$–$10^{-11}$ M). At each detection, the previous analyte solution was first removed from chamber by pump (Kylin-Bell Lab Instruments), followed by injection of current analyte solution, and then the signal collection was performed after the response time of ~2 minutes for the sensor response to reach the dynamical equilibrium. During $Hg^{2+}$ detection in any one of the prepared $Hg^{2+}$ concentrations, the collection of data was performed for 5 times, and the average value and y-axis error bar were taken from the five determined resonance angle shifts.

**Solution preparation.** Deionized water was first prepared from distilled water via an ion exchange water filter system, and then it was filtered one more time via an ultrapure water filter system to give the final ultrapure water, which is used in all ion solution preparations. The fresh as-prepared ultrapure water has an initial pH value of ~7, but is observed to lower and stabilize at ~6.1 several hours after preparation. The $Hg^{2+}$ solutions of different concentrations were prepared via 10-fold serial dilution of the mother liquid ($10^{-2}$ M), which is prepared from the purchased high-purity $Hg^{2+}$ solution ($Hg(NO_3)_2$, ThermoFisher), with ultrapure water and without additional acidification. For all the prepared $Hg^{2+}$ solutions, the pH values were carefully checked before any $Hg^{2+}$ detection, and they were found to be almost a constant value of ~6.1. The error bar for $Hg^{2+}$ concentration ($\Delta D$) is estimated using the expression:

$$\Delta D = \pm \sqrt{\left(\frac{v}{V+v}\right)^2 (\Delta d)^2 + \left(\frac{dv}{(V+v)^2}\right)^2 (\Delta V)^2 + \left(\frac{dV}{(V+v)^2}\right)^2 (\Delta v)^2}, \quad (1)$$

where $\Delta V$ and $\Delta v$ are the errors in volume measurements of ultrapure water and to-be-diluted $Hg^{2+}$ solution, respectively, and $\Delta d$ is the concentration error of to-be-diluted $Hg^{2+}$ solution. The error bar ($\Delta D$) of the $10^{-18}$ M $Hg^{2+}$ solution was estimated to be $\pm 0.1 \times 10^{-18}$ M and it is noted in the caption of Fig. 2b as the x-error bar for the concentration of $10^{-18}$ M.

**DFT calculation details.** All first principles calculations were performed by using PWmat code[50,51] with the scalar-relativistic (SG15) norm-conserving pseudopotential[52]. To mimic grain boundaries in $WS_2$, we employed the supercell approach with a vacuum thickness of 20 Å. Three different types of GBs were considered. All GBs were relaxed until the force on each atom was less than 0.03 eV/Å with Γ centered Monkhorst-Pack k-point mesh and an energy cutoff of 350 eV. The adsorption energies of Hg on GBs are evaluated by using $E_{ads} = E_{slab}^{ads} - E^{ads} - E_{slab}$, where $E_{slab}^{ads}$, $E^{ads}$, and $E_{slab}$ are the energies of calculated adsorbates-slab, adsorbates in gas phase and pure slab[53].

## Data availability

The data that support the findings of this study are available from the corresponding author upon reasonable request.

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

## Acknowledgements
We acknowledge the financial support from the National Natural Science Foundation of China (51732010), the Natural Science Foundation for Excellent Young Scholars of Hebei Province (E2020203140), and "100 Talents Project" of Hebei Province (E2020050016).

## Author contributions
Z.Y.L., T.Y.X., A.M.N. and Y.J.G. proposed, designed, and supervised the project. L.X.L. and K.Y. made equal contribution to the work. They performed the CVD growth of GB-rich polycrystalline 1L W(Mo)S$_2$ films, film characterizations of OM, FL, Raman, PL, and AFM, fabrication of SPR sensors and sensor characterizations. A.M.N. did TEM measurements. C.Q.L. and Y.C.C. did DFT calculations. Z.Y.J. and J.Y.X. supervised the CVD growth. C.P.M., B.C.W., F.S.W. and K.Z. supervised the OM, FL, Raman, PL, and AFM measurements. L.X.L. wrote the paper. Z.S.Z., Y.J.T., and all the other authors were involved in discussions and comments on the collected data and manuscript.

## Competing interests
The authors declare no competing interests.
