## [Peer Review File · Nature Communications]

REVIEWER COMMENTS

Reviewer #1 (Remarks to the Author):

This study reports 2D material-based SPR sensors for Hg²⁺ detection. Grain boundaries in 2D material (WS₂) were artificially created to enhance the detection ability, achieving an LOD down to aM level. The Hg²⁺ sensing performance is high and may attract some research attention within this field. However, I have to say that the overall quality of the manuscript needs to be greatly improved. Authors put a lot of focus on the material synthesis and characterizations, but some key features of the sensor were not investigated or studied.

Specific comments and suggestions:

1. The title of "2D material" should be more specific, and the target analyte could also be included.
2. Keywords, the "biological and chemical sensors" doesn't match well with Hg²⁺ detection.
3. Too much discussions were given on the synthesis of 2D materials, and the importance and significance of this work were not fully discussed. Biological and chemical sensors are a broad research area, you should make it clear what kind of sensor you are working on at the very beginning of the article, and highlight the breakthroughs and novelty of this type of grain-boundary-rich 2D material-based sensor.
4. The 2D material applied in the sensor device is WS₂; however, no words about WS₂ can be found in introduction. Should discuss why WS₂ was used as the sensing platform. What are the superiorities of WS₂ compared to other 2D materials or other TMDs?
5. From figure 1a-b, the AFM images can hardly demonstrate that it is 1L WS₂.
6. Is it possible to control the density of GBs in WS₂? How the density and distribution of GBs within the 2D WS₂ influence the sensing performance of the sensor?
7. This sensor shows a good selectivity, as show in Figure 4a, but the reason for the high selectivity was not fully understood or studied. A non-negligible sensing response of this sensor for Pb²⁺ was also found compared with that of Hg²⁺. How about the DFT and XPS characterizations for other ions on WS₂? How a coexisting of Pb²⁺ and Hg²⁺ in the sample makes an influence on the detection result? Suggest add the selectivity results with mixed ion solutions?
8. The detection of Hg²⁺ on GBs-rich WS₂ SPR sensor is superior to most of the reported works in terms of LOD (Figure 4b). However, the ppb or sub ppb level detection is good enough for Hg²⁺ detection. I am thinking that an LOD of 1 aM is not needed for Hg²⁺ detection. What type of real samples you are trying to test? And what is the common level of Hg²⁺ in that sample?
9. The accuracy, stability, repeatability, and reusability are also critical factors determining the performance of a sensor. However, all these performances were not investigated in this study.
10. The language of this manuscript needs significant improvements. There are a lot misnomers and inaccurate expressions throughout this manuscript. I highly recommend polish the writing with a journal editor.

See examples:

In the title, there is no such expression as "attomolar-ability", which is unidiomatic.

In abstract, there is an overuse of adjectives, dense and ponderous sentence, which makes it hard to get the key points. Suggest use short sentences. This problem can also be found in many other parts of the manuscript, such as "the selective and sensitive low-limit detection of...".

Line 30, "we propose achieving..."

Line 32, the use of "ubiquitous" is inappropriate here.

Line 36, the use of "substantial", it is not a matching adjective for "sensitivity enhancement".

Line 36-37, "...Hg²⁺ detection down to trace attomolar-level quantification"

Reviewer #2 (Remarks to the Author):

The major aim of this paper is to provide an application of chemical vapor deposition (CVD) grown polycrystalline $1L Mo(W)S_2$ films as sensors with high sensitivity, selectivity and ultra-low limit of detection (at attomolar level). Authors claim that this type of sensors can be used as biochemical sensors but, as a proof-of-concept, they select the applications in divalent mercury ions analysis, which is not a typical biochemical analysis (mercury usually is not found in living organisms).

The work is novel and will be of interest to others in the community especially in the field of Analytical Chemistry.

The achieved minimum detection limit in divalent mercury ions detection is impressive (attomolar sensitivity, i.e. $10^{-18} M$, i.e. much lower (about five orders of magnitude) than the existing mercury ions detection limits (please see Fig. 4), but my opinion is that the manuscript cannot be accepted, and further work is needed in order to justify a resubmission for the following reasons.

Authors claim (page 12 last line) that the sensor displays a wide detectable dynamic range from $10^{-11} M$ to $10^{-18} M$, covering 7 orders of magnitude. But results given in Fig. 2 show that concentrations in the range from $10^{-11} M$ to $10^{-13} M$ give similar results (angle shift), taking into account the error bars. Additionally, and for the same reason, concentrations from $10^{-13} M$ to $10^{-14} M$ seems difficult to be distinguished

Authors claim that the sensor "is highly selective towards Hg^{2+} detection" (page 16, last line) which is "one of the crucial performance criteria" (page 16, line 293). In order to prove this statement, they compare the angle shift results of mercury ions with the results of other 8 elements (ions) of similar concentration ($10^{-12} M$) and they conclude that "the value of $\Delta\theta$ induced by Hg^{2+} adsorption is at least $\sim 3X$ higher than those produced by other ions, indicating that the GB-rich WS_2 sensor is highly selective towards Hg^{2+} detection" (page 16, lines 298-300). Mercury ions are really $3X$ higher than those produced by Pb ions but this is not enough to accept the method as "highly sensitive"; e.g. in the case that Pb ions have 10 times higher concentration than mercury ions they will give a much stronger angle shift than Hg ions and will interfere Hg detection (it is not possible to estimate if the angle shift comes from Hg, Pb or any other ion).

Additionally, the use of the same concentration value ($10^{-12} M$) for all the examined ions is not acceptable as (in real world) concentrations of Mg, Fe, Zn, Ca (calcium was not examined) etc. usually are more than 3-6 orders of magnitude higher than Hg ions concentrations.

Another problem is the lack of experimental information for the way of mercury solutions (concentration standards) preparation. Although for "ordinary" concentrations we can accept that standard solutions production (e.g. by dilution) does not add any significant error, if we produce standards of the order of $10^{-18} M$ we may insert significant error that has to be estimated and noticed as x-axis error bars.

A definition of the way of y-axis error bars estimation is also needed.

Authors write that they use "ultra pure water" but in concentrations up to $10^{-18} M$ the "ultra pure water" could be not so pure (for mercury, as well as for other ions). Did they examine this case?

There is not any information about any pH arrangement. If pH was 7 (i.e. for ultra pure water at 25 deg. Celsius) and not acidified, the mercury ions can be adsorbed on the walls of the (glass ?) vessel. If they had acidified the water solutions, there is the possibility of interferences from the introduced acid. Did they examine these parameters? In general, much more experimental information is needed (e.g. time of measurement, water volume effect, temperature etc.) in the supplementary part, to give the ability to other researchers to reproduce the work.

Fig. 4 parts a and b shall be separated to two figures as they give different information

Sincerely yours,

Nikolaos Kallithrakas-Kontos
Professor of Analytical Chemistry

Reviewer #3 (Remarks to the Author):

This manuscript reports on the fabrication of a hybrid surface plasmon resonance (SPR) sensor based on a gold film covered by high-density grain boundaries (GBs) monolayer (1L) WS₂ film for mercury ion (Hg²⁺) detection. The authors demonstrated the importance of the GBs as active sites for the selective and highly sensitive detection of Hg²⁺, reaching an astonishing limit of detection in the attomolar range. However there is not much novelty in this work because the CVD growth of wafer-scale polycrystalline 1L W(Mo)S₂ films has been previously published by the same authors. The main figures of merit of the present work are the selectivity and the superior sensitivity as compared to other reported substrates. Overall, I would recommend the publication of this contribution only when the following issues are addressed.

The authors reported the resonance angle shifts as a function of Hg²⁺ concentration showing that they can effectively detect Hg²⁺ in aqueous solutions at concentrations ranging from 10⁻¹¹ to 10⁻¹⁸ M (Figure 2). Could the authors comment if and how Hg²⁺ quantification is possible from these data? It appears that figure 2b reveals three different regions: i) A saturated region, where the SPR response is independent of the Hg²⁺ concentration ii) a quantification region where the response seems to be linear with the Hg²⁺ concentration and iii) a detection region where there is a response to Hg²⁺ but can not be quantified.

The authors have shown that their system is selective towards Hg²⁺ in the presence of common interfering ions such as Pb²⁺, Mn²⁺, Cu²⁺, Fe³⁺, Zn²⁺, Co²⁺, Cr³⁺ and Mg²⁺. However, the selectivity of the presented sensor is based on the Hg-S bond formation and hence any other species than can interact with sulfur will also respond (e.g. Ag⁺ or As³⁺). The authors should clarify that in the manuscript.

Could the authors comment on the response time of their sensors? Is the signal collected after the immediate exposure of the substrate to the Hg²⁺ solution or is there any delay time? If so, the authors should indicate it.

The assessment of a real matrix (e.g. tap water) for Hg²⁺ sensing would be of high interest to demonstrate that the robustness of sensor's performance.

Minor issues:

Figure S4 is cited before S3 in the manuscript. The numbering should be reversed.

Did the authors test the response of their sensor with higher concentrations (>10⁻¹¹ M) of Hg²⁺?

Or above this concentration the sensor's response is saturated?

Did the authors test the recyclability of their substrates?

Firstly, we would like to appreciate the Reviewers' comments, based on which our revised manuscript has been significantly improved. The revisions based on the reviewers' comments are highlighted in blue in the revised manuscript.

REVIEWER COMMENTS AND OUR CORRESPONDING REPLIES

Reviewer #1 (Remarks to the Author):

This study reports 2D material-based SPR sensors for Hg²⁺ detection. Grain boundaries in 2D material (WS₂) were artificially created to enhance the detection ability, achieving an LOD down to aM level. The Hg²⁺ sensing performance is high and may attract some research attention within this field. However, I have to say that the overall quality of the manuscript needs to be greatly improved. Authors put a lot of focus on the material synthesis and characterizations, but some key features of the sensor were not investigated or studied.

Reply:

In response to the Reviewers' comments, we have made significant improvements to overall quality of the revised manuscript. In the revised introduction, most of the discussions about synthesis and characterizations of 2D materials have been moved to SI, and main attention has been paid to the GB-rich polycrystalline 1L WS₂ film, the GB-based sensor and the critical features of sensor performance. Additional investigations have been carried out on critical features of the sensor performance, including accuracy, stability, repeatability, recyclability, and interference studies. All the additional data have been discussed and added in the revised manuscript.

Specific comments and suggestions:

1. The title of “2D material” should be more specific, and the target analyte could also be included.

Reply:

We thank the reviewer for the constructive suggestions. In response to the comment, the title has been changed to “Grain-boundary-rich polycrystalline monolayer WS₂ film for attomolar-level Hg²⁺ sensors” in the revised manuscript.”

2. Keywords, the “biological and chemical sensors” doesn’t match well with Hg²⁺ detection.

Reply:

We thank the reviewer for pointing out this issue. In response, the Keyword “biological and chemical sensors” has been replaced with “metal ion sensors” in the revised manuscript.

3. Too much discussions were given on the synthesis of 2D materials, and the importance and significance of this work were not fully discussed. Biological and chemical sensors are a broad research area, you should make it clear what kind of sensor you are working on at the very beginning of the article, and highlight the breakthroughs and novelty of this type of grain-boundary-rich 2D material-based sensor.

Reply:

We thank the reviewer for the constructive suggestions. In response to the comment, we have made major changes to the introduction in the revised manuscript. We have deleted most of the discussions about 2D

material synthesis and focused on the importance and significance of the present work. In the revised version, we have directly pointed out the fabrication of SPR sensor by using GB-rich polycrystalline 1L WS₂ films as sensing materials for Hg²⁺ detection, and we have highlighted the breakthroughs and novelty of the GB-based SPR sensor. We have pointed out the main reasons and advantages for selecting WS₂ film as sensing material in fabrication of SPR sensor for Hg²⁺ detection.

4. The 2D material applied in the sensor device is WS₂; however, no words about WS₂ can be found in introduction. Should discuss why WS₂ was used as the sensing platform. What are the superiorities of WS₂ compared to other 2D materials or other TMDs?

Reply:

We thank the reviewer for pointing out these issues. In response to the comment, we have focused our discussions to WS₂ in the revised introduction and pointed out the main reasons for selecting WS₂ film as sensing material in fabrication of SPR sensor for Hg²⁺ detection. The main reasons and advantages for our choice of GB-rich 1L WS₂ film as the sensor material are shown below. 1. Controllable CVD growth of centimeter-scale GB-rich 1L WS₂ film with nanoscale grains in a narrow distribution around an average size of ~40 nm can be achieved. For most of the other TMDs, the large-area growth of one or few-layer film is still a big challenge up to now. 2. Desirable active S sites of huge amounts on high-density 1L WS₂ GBs can be used for preferential or selective adsorption of Hg²⁺. 3. WS₂ shows excellent stability in air or solution, which is beneficial for application of the GB-based sensor in complex solutions and long shelf life without strict requirements for storage

conditions. For most of 2D layered materials such as phosphorene and NbS₂, the environmental stability is a big problem for their potential applications.

5. From figure 1a-b, the AFM images can hardly demonstrate that it is 1L WS₂.

Reply:

We thank the reviewer for pointing out this issue. In response to the comment, the AFM height profile has been moved to Fig. 1a as inset in the revised manuscript for distinct demonstration of the 1L nature.

6. Is it possible to control the density of GBs in WS₂? How the density and distribution of GBs within the 2D WS₂ influence the sensing performance of the sensor?

Reply:

We thank the reviewer for this interesting question. Up to now, the authors have spent several years to exploit the growth parameters for CVD synthesis of 1L and few-layer WS₂ crystals and films. By using the H₂S gas as S source in CVD synthesis of GB-rich polycrystalline 1L WS₂ film on SiO₂ (300 nm)/Si substrate, we have been able to control the growth of 1L WS₂ grains in the nanoscale sizes for the hugely increased density of GBs. In this case, the distribution of grains is narrow (between 20 to 55 nm) and the average size is ~40 nm. Single crystalline 1L WS₂ with grain size of tens of μm grown by using S powder is applied as control group. As shown in Figure 2a in main text, single crystalline 1L WS₂ shows much worse sensitivity. Apparently, the sensor device will

have better performance with higher density of GBs and narrower distribution of the grain size will result in more repeatable results.

In the revised manuscript, we have added detailed discussion about the essential role of nanoscale grains or high density of GBs in ultra-sensitive response of the GB-based SPR sensor in Hg^{2+} detection (Supplementary discussion). The smaller sizes of 1L WS_2 grains imply the higher density of GBs, which is able to provide more structural defects as active sites for capture of analyte and thus the higher sensing performance of the GB-based sensor. Sharply increased density of GBs can only be achieved with the growth of nanoscale grains (Supplementary Fig. 1). The microscale or larger 1L WS_2 grains (Supplementary Fig. 3) are actually not desirable for the great enhancement of sensor performance, because the larger grain sizes mean the lower density of GBs. As for GB distribution in GB-rich polycrystalline 1L WS_2 film, because grain size distribution is also constrained in the nanoscale range, and thus, only minute local fluctuation exists in the density of GBs across the film. Considering the 170 mm spot size of detection light, which is at least 3 orders larger than nanoscale grains, the density and distribution of GBs within the detection range is thus pretty uniform for different areas, guaranteeing consistent sensing performances of the GB-based SPR sensor from different areas of the film (Supplementary Fig. 2). The large 1L WS_2 grains and the wide size distribution, however, inevitably lead to the large local fluctuation in the density or distribution of GBs across the film. The large local fluctuation in density or distribution of GBs is found to induce the inconsistent sensor performances in Hg^{2+} detection from the different areas of the film, as shown in Supplementary Fig. 4.

Supplementary Fig. 1. SEM image of the initially grown 1L WS_2 nanocrystals of equilateral triangular shape and high surface density on a SiO_2 (300 nm)/Si substrate, which were intercepted by intentionally stopping the CVD growth process of the GB-rich polycrystalline 1L WS_2 film. Inset on top is the high-magnification SEM image, and inset on bottom gives the calculated density D_{GB} of GBs as a function of lateral length ℓ .

Supplementary Fig. 2. Determined resonance angle shifts of $\Delta\theta$ from 5 different areas on the CVD-grown GB-rich polycrystalline 1L WS_2 film as sensing material in the GB-based SPR sensor. The consistent values of $\Delta\theta$ indicate the desirable growth of nanoscale grains for consistent detection of Hg^{2+} from the different locations of the film because the spot size of detection light (170 μm in diameter) is much greater than nanoscale grain dimensions.

Supplementary Fig. 3. (a) OM image of the microscale 1L WS₂ crystals grown on SiO₂ (300 nm)/Si substrate, which were intercepted by intentionally stopping the CVD growth process of polycrystalline 1L WS₂ film in the early stage. The S powder was used as S source in the CVD growth. (b) OM (top) and corresponding FL (bottom) images of the CVD-grown polycrystalline 1L WS₂ film. In the FL image, the FL-enhanced stripes come from the GBs.⁷ As revealed in the FL image, the 1L WS₂ grains lie in a large size distribution from several tens to ~200 μm, and thus the quite non-uniform distribution of GBs leads to large local fluctuation in the density of GBs across the film.

Supplementary Fig. 4. Sensing performances of SPR sensor based on 1L WS₂ film with microscale grains. The sensor was fabricated with the CVD-grown polycrystalline 1L WS₂ film as shown in Fig. S3b. **(a)** SPR spectra taken from two different areas on the polycrystalline 1L WS₂ film as sensing material. **(b)** Resonance angle shifts of $\Delta\theta$ as a function of Hg²⁺ concentration, which were determined from the SPR spectra in **(a)**. Obviously, different areas on the film generate the inconsistent sensor responses due to large fluctuation in the density of GBs across the film.

7. This sensor shows a good selectivity, as shown in Figure 4a, but the reason for the high selectivity was not fully understood or studied. A non-negligible sensing response of this sensor for Pb²⁺ was also found compared with that of Hg²⁺. How about the DFT and XPS characterizations for other ions on WS₂? How a coexisting of Pb²⁺ and Hg²⁺ in the sample makes an influence on the detection result? Suggest adding the selectivity results with mixed ion solutions?

Reply:

We thank the reviewer for the constructive suggestions. In response to the corresponding comments of all three Reviewers, additional investigations have been done to check the possible interference to the sensor response from any other coexisting metal ion in Hg^{2+} detection. In consideration of the observed non-negligible sensor response to Pb^{2+} (Ag^+) adsorption, a series of mixed solutions with the coexisting Hg^{2+} and Pb^{2+} (Ag^+) ions have been prepared. In the prepared mixed solutions, the Hg^{2+} concentration is fixed at 10^{-15} M, while the Pb^{2+} (Ag^+) concentration is changed from 10^{-16} to 10^{-14} M. Compared with the sensor response to the 10^{-15} M Hg^{2+} solution, no obvious interference to the sensor response has been observed from the coexisting Pb^{2+} (Ag^+) ions in the mixed solutions, even if the concentration of Pb^{2+} (Ag^+) is increased by 10 times higher than that of Hg^{2+} ions. After the detection in the mixed solutions of Hg^{2+} and Pb^{2+} ions, the XPS examination of sensor material exhibits no sign of Pb^{2+} in the Pb^{2+} 4f core level spectrum (Supplementary Fig. 13). This phenomenon can be attributed to the preferential adsorption of Hg^{2+} on GBs. In response to the Reviewer's suggestion, we have provided the selectivity results in the mixed solutions of Hg^{2+} and Pb^{2+} (Ag^+) ions as Fig. 4b and c in the revised manuscript, and we have given the discussion and our understanding about the reason for high selectivity of GB-based sensor response to Hg^{2+} adsorption on the basis of hard-soft-acid-base (HSAB) theory. Just DFT calculations of Hg^{2+} adsorption were conducted to demonstrate the preferential absorption of Hg^{2+} ions on the GBs instead of the

defect-free areas in the GB-rich 1L WS₂ film. The experimental interference results have indicated the much more selective absorption of Hg²⁺ ions on the GBs than the other tested metal ions. No DFT calculation was performed on the other ions.

Supplementary Fig. 13. XPS spectra of Hg 4f (a) Pb 4f (b) core levels taken from the GB-rich polycrystalline 1L WS₂ film as sensing material after ion detection in the mixed Hg²⁺ and Pb²⁺ solutions, in which Hg²⁺ concentration is 10⁻¹⁵ M and Pb²⁺ concentration is 10⁻¹⁴ M.

Fig. 4. Selectivity of Hg^{2+} detection in mixed solutions of coexisting Hg^{2+} and Pb^{2+} ions **(b)** and of coexisting Hg^{2+} and Ag^{+} ions **(c)**. In the mixed solutions, Hg^{2+} concentration is fixed at 10^{-15} M, and Pb^{2+} (Ag^{+}) is changed from 10^{-16} to 10^{-14} M. $\Delta\theta$ is the resonance angle shift.

8. The detection of Hg^{2+} on GBs-rich WS_2 SPR sensor is superior to most of the reported works in terms of LOD (Figure 4b). However, the ppb or sub ppb level detection is good enough for Hg^{2+} detection. I am thinking that an LOD of 1 aM is not needed for Hg^{2+} detection. What type of real samples you are trying to test? And what is the common level of Hg^{2+} in that sample?

Reply:

We thank the reviewer for this interesting question. In the submitted work, we proposed the application potential of GB-rich polycrystalline one or few-layer film of 2D layered materials as ultra-sensitive sensing materials in SPR sensors. As an example, we fabricated the SPR sensor with GB-rich polycrystalline 1L WS_2 film as sensor material for Hg^{2+} detection, and we demonstrated that the sensor's sensitivity can be hugely enhanced to a LOD of 1 aM level due to the large amounts of defects or active sites on high-density GBs. We agree that a LOD of 1 aM is not required in practical application of Hg^{2+} detection. From a fundamental research perspective, it is of significance to find that the GB-based SPR sensor is able to reach a 1 aM-level sensitivity in Hg^{2+}

detection, because it demonstrates the great potential of GBs of 2D materials in fabrication of ultra-sensitive ion sensors. The GBs-rich WS₂ provides enough active sites to adsorb Hg²⁺ ions and the ultrathin thickness of 2D materials makes them very sensitive to the adsorbed species, which will be complementary to the sensors based on traditional materials. We believe that our work will directly stimulate the exploitation of GB-rich polycrystalline one or few-layer film of any other layered metal sulphide as sensing material for fabrication of highly sensitive ion sensors. More broadly, our work will also inspire the exploitation of GB-rich polycrystalline one or few-layer film of any other layered material as sensor materials for detection of wide types of analytes. We added these discussions to the main text.

9. The accuracy, stability, repeatability, and reusability are also critical factors determining the performance of a sensor. However, all these performances were not investigated in this study.

Reply:

We thank the reviewer for the constructive suggestions. In response to this comment, additional investigations have been performed on accuracy, stability, repeatability, and reusability of the GB-based SPR sensors.

For the detection accuracy of a SPR sensor, it is defined as $Da=1/W_{FWHM}$, where W_{FWHM} is the full width at half maximum of SPR spectrum. For the SPR sensors based on GB-rich 1L WS₂ film, the accuracy of $Da\sim 0.4$ was achieved, which is better than that of the graphene-based SPR sensor (~ 0.2 , 680 nm laser light) (see Supplementary discussion).

As for the repeatability, a series of GB-based SPR sensors were fabricated with the GB-rich polycrystalline 1L WS₂ films grown from 15

batches on different dates, and their responses to the 10^{-15} M Hg^{2+} solution were examined. As shown in Supplementary Fig. S14b, the determined resonance angle shifts of $\Delta\theta$ exhibit the pretty good consistency and thus the excellent repeatability.

In addition to the environmental stability, the GB-based SPR sensor also involves the stability in the detected solutions. To check the stability in solutions, the sensor response to 10^{-15} M Hg^{2+} solution was monitored via Kinetic curve for more than 2 hours, as shown in Supplementary Fig. 15 in the revised manuscript. After reaching the dynamical equilibrium, the response signal becomes quite steady for at least 2 hours, and no observation of any abrupt change exhibits the good sensor stability in the detected solution. To test the environmental stability, for a GB-based sensor after half-year exposure under ambient conditions, its responses to the Hg^{2+} solutions from 10^{-18} to 10^{-11} M were investigated. Before the investigations, the sensor was treated at 260°C for 10 minutes under the protection of mixed H_2 (10%) and Ar gas to remove the adsorbed contaminants on the GBs. As shown in Supplementary Fig. 16, after 6-month exposure to air, the sensor still exhibits comparable sensor performance to that of the fresh as-grown GB-rich 1L WS_2 film. These observations indicate the excellent environmental stability of GB-based sensor.

In consideration of the Hg^{2+} adsorption via S-Hg bond formation on GBs, the GB-rich polycrystalline 1L WS_2 film as sensing material cannot be recycled after detection, but the reusability of Au film as substrate can be realized just by ultrasonic removal of 1L WS_2 film in water since the weak attachment of 1L WS_2 film on Au film via van der Waals forces.

Supplementary Fig. 14. Resonance angle shifts of $\Delta\theta$, which were determined from the responses of 15 GB-based SPR sensors to the 10^{-15} M Hg^{2+} solution. These sensors were fabricated with GB-rich polycrystalline 1L WS_2 films from 15 batches grown at different dates. The consistency in values of $\Delta\theta$ and small standard deviation ($\Delta\theta = 89.28 \pm 2.89$ mdeg) indicates the high repeatability.

Supplementary Fig. 15. Response of one GB-based SPR sensor to 10^{-15} M Hg^{2+} solution monitored by the Kinetic curve for more than 2 hours. Inset is the

amplified Kinetic curve to show the initial response. The arrow-pointed jump is induced by injection of Hg^{2+} solution.

Supplementary Fig. 16. Environmental stability of GB-rich 1L WS_2 film. SPR spectra of freshly grown and 6-month exposed GB-rich 1L WS_2 film at Hg^{2+} concentrations ranging from 10^{-18} M to 10^{-11} M

10. The language of this manuscript needs significant improvements. There are a lot misnomers and inaccurate expressions throughout this manuscript. I highly recommend polish the writing with a journal editor.

See examples:

In the title, there is no such expression as “attomolar-ability”, which is unidiomatic.

In abstract, there is an overuse of adjectives, dense and ponderous sentence, which makes it hard to get the key points. Suggest use short sentences. This problem can also be found in many other parts of the manuscript, such as “the selective and sensitive low-limit detection of...”.

Line 30, “we propose achieving...”

Line 32, the use of “ubiquitous” is inappropriate here.

Line 36, the use of “substantial”, it is not a matching adjective for “sensitivity enhancement”.

Line 36-37, “...Hg²⁺ detection down to trace attomolar-level quantification”

Reply:

We thank the reviewer for pointing out the language issues. The “attomolar-ability” in the title has been corrected as “attomolar-level” in the revised manuscript. The language of the main text has been carefully polished, and the overuse of adjectives, dense and ponderous sentences, misnomers and inaccurate expressions, as pointed out by the reviewer, have been corrected. Especially, the abstract is re-written as below:

Emerging two-dimensional (2D) layered materials have been attracting great attention as sensing materials for next-generation high-performance biological and chemical sensors. The sensor performance of 2D materials is strongly dependent on the structural defects as indispensable active sites for analyte adsorption. However, controllable defect engineering in 2D materials is a big challenge. In the present work, we propose exploitation of controllably grown polycrystalline films of 2D layered materials with high-density grain boundaries (GBs) for design of ultra-sensitive ion sensors, where abundant structural defects on GBs act as favorable active sites for ion adsorption. As a proof-of-concept, our fabricated surface plasmon resonance sensors with GB-rich polycrystalline monolayer WS₂ films have exhibited high selectivity and superior attomolar-level sensitivity in

Hg²⁺ detection owing to high-density GBs. This work provides a promising avenue for design of ultra-sensitive sensors based on GB-rich 2D layered materials.

Reviewer #2 (Remarks to the Author):

The major aim of this paper is to provide an application of chemical vapor deposition (CVD) grown polycrystalline 1L Mo(W)S₂ films as sensors with high sensitivity, selectivity and ultra-low limit of detection (at attomolar level). Authors claim that this type of sensors can be used as biochemical sensors but, as a proof-of-concept, they select the applications in divalent mercury ions analysis, which is not a typical biochemical analysis (mercury usually is not found in living organisms).

Reply:

We thank the reviewer for pointing out this issue. In response, “biochemical analysis” has been corrected as “metal ion analysis” in the revised manuscript.

The work is novel and will be of interest to others in the community especially in the field of Analytical Chemistry.

The achieved minimum detection limit in divalent mercury ions detection is impressive (attomolar sensitivity, i.e. 10⁻¹⁸M, i.e. much lower (about five orders of magnitude) than the existing mercury ions detection limits (please see Fig. 4), but my opinion is that the manuscript cannot be accepted, and further work is needed in order to justify a resubmission for the following reasons.

Authors claim (page 12 last line) that the sensor displays a wide detectable dynamic range from 10⁻¹¹M to 10⁻¹⁸M, covering 7 orders of magnitude. But results given in Fig. 2 show that concentrations in the range from 10⁻¹¹M to 10⁻¹³M give similar results (angle shift), taking into account the error bars. Additionally, and for the same reason, concentrations from 10⁻¹³M to 10⁻¹⁴M seems difficult to be distinguished

Reply:

We thank the reviewer for pointing out this issue. In the revised manuscript, the corresponding description about Fig. 2 has been modified as “Notably, compared with the 1L WS₂ single crystal based sensor, the GB-based sensor displays a much wider detectable range of Hg²⁺ concentration than that of 1L WS₂ single crystal. For the GB-based SPR sensor, the most sensitive response occurs in the concentration range from 10⁻¹⁷ to 10⁻¹³ M. Below 10⁻¹⁷ M, the sensor response becomes weaker and exact Hg²⁺ quantification is not easy, but, semi-quantitative analysis regarding the order of detected Hg²⁺ concentration is still achievable from the SPR resonance angle shift, as it is still discernible even at the attomolar-level concentration (13 milli-degree for 10⁻¹⁸ M). Above 10⁻¹³ M, the response tends to be saturated. The drastically enhanced performance of GB-based sensor reveals the significant role of high-surface-density GBs in sensor performance.”.

Authors claim that the sensor "is highly selective towards Hg²⁺ detection" (page 16, last line) which is "one of the crucial performance criteria" (page 16, line 293). In order to prove this statement, they compare the angle shift results of mercury ions with the results of other 8 elements (ions) of similar concentration (10⁻¹²M) and they conclude that "the value of $\Delta\theta$ induced by Hg²⁺ adsorption is at least ~ 3X higher than those produced by other ions, indicating that the GB-rich WS₂ sensor is highly selective towards Hg²⁺ detection" (page 16, lines 298-300). Mercury ions are really 3X higher than those produced by Pb ions but this is not enough to accept the method as "highly sensitive"; e.g. in the case that Pb ions have 10 times higher concentration than mercury ions they will give a much stronger angle shift than Hg ions and will interfere Hg detection (it is not possible to estimate if the angle shift comes from Hg, Pb or any other ion).

Reply:

We thank the reviewer for this interesting question. We agree with the Reviewer that the interference to sensor response from the coexisting Pb^{2+} ions in Hg^{2+} detection might be an issue. In response to the comment and also the corresponding ones of the other two Reviewers, additional investigations have been done to check the possible interference to the sensor response from any other coexisting metal ion in Hg^{2+} detection. Additional examination on the sensor response to the 10^{-12} M Ag^+ solution also shows non-negligible response, a little stronger than that to the 10^{-12} M Pb^{2+} solution. In consideration of the observed non-negligible sensor response to Pb^{2+} (Ag^+) adsorption, a series of mixed solutions with the coexisting Hg^{2+} and Pb^{2+} (Ag^+) ions have been prepared for examination on the possible interference from Pb^{2+} (Ag^+). In the prepared mixed solutions, the Hg^{2+} concentration is fixed at 10^{-15} M, while the Pb^{2+} (Ag^+) concentration is changed from 10^{-16} to 10^{-14} M. Compared with the sensor response to the 10^{-15} M Hg^{2+} solution, no obvious interference to the sensor response from the coexisting Pb^{2+} (Ag^+) ions in the mixed solutions has been observed, even if the concentration of Pb^{2+} (Ag^+) is increased by 10 times higher than that of Hg^{2+} ions. After the detection in the mixed solutions of Hg^{2+} and Pb^{2+} ions, the XPS examination of sensor material exhibits no sign of Pb^{2+} in the Pb 4f core level spectrum. In response to the Reviewers' suggestion, the selectivity results obtained in the mixed solutions have been added as Fig. 4b and c in the revised manuscript.

Fig. 4. Selectivity of Hg²⁺ detection in mixed solutions of coexisting Hg²⁺ and Pb²⁺ ions **(b)** and of coexisting Hg²⁺ and Ag⁺ ions **(c)**. In the mixed solutions, Hg²⁺ concentration is fixed at 10^{-15} M, and Pb²⁺ (Ag⁺) is changed from 10^{-16} to 10^{-14} M. $\Delta\theta$ is the resonance angle shift.

Additionally, the use of the same concentration value (10^{-12} M) for all the examined ions is not acceptable as (in real world) concentrations of Mg, Fe, Zn, Ca (calcium was not examined) etc. usually are more than 3-6 orders of magnitude higher than Hg ions concentrations.

Reply:

We thank the reviewer for pointing out this issue. In response to the Reviewer's concern, additional investigations have been performed to examine the sensor responses to the 10^{-6} M solutions of Mg²⁺, Fe²⁺, Zn²⁺ and Ca²⁺. The determined resonance angle shifts of $\Delta\theta$ have been provided in Supplementary Fig. 12 in the revised manuscript. Compared with the negligible $\Delta\theta$ at the 10^{-12} M concentration of Fe³⁺, Zn²⁺, Mg²⁺ and Ca²⁺ ions, although a stronger sensor response or larger $\Delta\theta$ is observed at the 10^{-6} M concentration, they are only at most one fifth of the Hg²⁺-produced $\Delta\theta$ at the concentration of 10^{-12} M, as shown in Supplementary Fig. 12. Furthermore, as revealed in the sensor response

to the mixed solution of 10^{-15} M Hg^{2+} and 10^{-14} M Pb^{2+} (Ag^+) ions, no obvious interference to the sensor response from Pb^{2+} (Ag^+) ions has been observed, even though the Pb^{2+} (Ag^+) concentration is 10 times higher than the Hg^{2+} concentration in the mixed solution. Based on these results, the negligible interference to the sensor response from Fe^{3+} , Zn^{2+} , Mg^{2+} and Ca^{2+} ions at the higher concentration would be also expected.

Supplementary Fig. 12. Determined resonance angle shifts of $\Delta\theta$ from the responses of GB-based SPR sensors to 10^{-6} M solutions of Fe^{3+} , Zn^{2+} , Mg^{2+} and Ca^{2+} ions. The value of $\Delta\theta = 129$ mdegree was obtained from the sensor response to 10^{-12} M Hg^{2+} solution, which is inserted for comparison. Compared with the negligible $\Delta\theta$ at the 10^{-12} M concentration of Fe^{3+} , Zn^{2+} , Mg^{2+} and Ca^{2+} ions, although a sensor response ($\Delta\theta$) is observed at the 10^{-6} M concentration, they are only one fifth of the Hg^{2+} -produced $\Delta\theta$ at the lower 10^{-12} M concentration. Thereby, the interference to the sensor response from Fe^{3+} , Zn^{2+} , Mg^{2+} and Ca^{2+} ions at the higher concentration would be also expected to be negligible.

Another problem is the lack of experimental information for the way of mercury solutions (concentration standards) preparation. Although for "ordinary" concentrations we can accept that standard solutions production (e.g. by dilution) does not add any significant error, if we produce standards of the order of 10^{-18} M we may insert significant error that has to be estimated and noticed as x-axis error bars.

A definition of the way of y-axis error bars estimation is also needed.

Reply:

We thank the reviewer for this interesting question. The Hg^{2+} solutions were prepared via 10-fold serial dilution of the mother liquid (10^{-2} M). The mother liquid is prepared from the purchased Hg^{2+} reagent ($\text{Hg}(\text{NO}_3)_2$, ThermoFisher), with ultra-pure water and without additional acidification. The error bar for Hg^{2+} concentration (D) is estimated via the expression of $\Delta D = \pm \sqrt{\left(\frac{v}{V+v}\right)^2 (\Delta d)^2 + \left(\frac{dV}{(V+v)^2}\right)^2 (\Delta v)^2 + \left(\frac{dv}{(V+v)^2}\right)^2 (\Delta V)^2}$, where ΔV and Δv are the errors in volume measurements of ultrapure water and to-be-diluted Hg^{2+} solution, respectively, and Δd is the concentration error of to-be-diluted Hg^{2+} solution. In preparation of the 10^{-18} M Hg^{2+} solution, the error bar of ΔD was estimated to be $\pm 0.1 \times 10^{-18}$ M, and it has been noted in the caption of Fig. 2b as the x-error bar for the concentration of 10^{-18} M. During Hg^{2+} detection in any one of the prepared Hg^{2+} concentrations, the collection of data was performed for 5 times, and the average value and y-axis error bar were taken from the five determined resonance angle shifts. In the revised manuscript, the definitions of x and y-axis error bars have been added in Supplementary methods.

Authors write that they use "ultra pure water" but in concentrations up to 10^{-18} M the "ultra pure water" could be not so pure (for mercury, as well as for other ions). Did they examine this case?

Reply:

We thank the reviewer for pointing out this issue. We understand the Reviewer's concern about the water purity. Actually, we had the similar concern about the concentration of as low as 10^{-18} M in the preparation process of Hg^{2+} solutions. During the solution preparation, we have paid careful attention to the preparation of ultra-pure water. To the best of our ability, we had taken some measures to guarantee the high purity of water used in the solution preparation. In the preparation process of ultrapure water, deionized water was first prepared from distilled water via an ion exchange water filter system, and then the obtained deionized water was filtered again via an ultrapure water filter system to give the finally used ultra-pure water. ICP analyses were performed to examine any possible presence of Hg^{2+} ions in the highly purified water, but no sign is observable. Moreover, the sensor was repeatedly tested in the highly purified water, and no observable change in the response signal (i.e resonance angle shift) was produced. This can prove that the concentration of Hg^{2+} , Ag^+ and Pb^{2+} in the ultra-pure water, if any, should be lower than 10^{-18} M, which will not influence the results in this manuscript.

There is not any information about any pH arrangement. If pH was 7 (i.e. for ultra pure water at 25 deg. Celsius) and not acidified, the mercury ions can be adsorbed on the walls of the (glass ?) vessel. If they had acidified the water solutions, there is the possibility of interferences from the introduced acid. Did they examine these parameters? In general, much more experimental

information is needed (e.g. time of measurement, water volume effect, temperature etc.) in the supplementary part, to give the ability to other researchers to reproduce the work.

Reply:

We thank the reviewer for pointing out this issue. We did examine the pH values of ultra-pure water and all the prepared Hg^{2+} solutions before the Hg^{2+} detection. For the fresh as-prepared ultra-pure water, the pH value was checked to be ~ 7 , but with time, the pH value was observed to decrease and stabilize at around 6. We did not do any intentional acidification during solution preparation. For all the prepared Hg^{2+} solutions from 10^{-18} to 10^{-9} M, the pH values are found to slightly fluctuate around 6.1, which have been added as Supplementary Fig. 17a in the revised manuscript. This can remove the concern that the SPR response is caused by pH change along with dilution. Also, the experimental result of concentration-dependent SPR response is strong evidence that the mercury ions adsorbed on the walls of the vessel should be a small ratio. Now we have intentionally acidified the water with nitric acid (HNO_3) to examine the impact of pH on sensor response. We have checked the sensor responses to a series of acidified water with the different pH values from 6.1 to 3, as shown in Supplementary Fig. 17. No obvious change is observed in the resonance angles for the pH values ranging from 6.1 to 5, implying the negligible interference to the sensor response from the solution pH of ~ 6.1 . For the pH value of below 5, however, the determined resonance angle is observed to be obviously shifted toward the smaller value, suggesting the non-negligible interference from introduced acid. Just as pointed out by the Reviewer in the comment, the pH value in the detected solution should be carefully checked in order to

avoid any interference to the sensor response from acid. In the revised manuscript, the detailed experimental information such as time of measurement, water volume effect, temperature has been also added in supplementary information, as pointed out by the Reviewer.

Supplementary Fig. 17. Sensor response to acidified water. (a) pH values in the prepared Hg^{2+} solutions, which slightly fluctuate around 6.1. Inset is the time-dependent pH value of ultrapure water after preparation. The pH value of as-prepared ultra-pure water is around 7, but it decreases with time to a stable value of ~6 under exposure to air. No acidification of water was done in the preparation of Hg^{2+} solutions. **(b)** SPR spectra of GB-based SPR sensor in a series of the acidified water. Acidification of ultra-pure water to the pH value ranging from 6.1 to 3 was performed by adding nitric acid (HNO_3). **(c)** Corresponding resonance angle shifts of $\Delta\theta$ extracted from the SPR spectra in **(b)**.

Fig. 4 parts a and b shall be separated to two figures as they give different information

Reply:

We thank the reviewer for the constructive suggestions. In response to the comment, Fig. 4 has been separated to two figures, i.e. Fig.4 and Fig. 5 in the revised manuscript, and additional selectivity results regarding the interference to the sensor response from coexisting Pb^{2+} (Ag^+) ions has been added in revised Fig.4.

Reviewer #3 (Remarks to the Author):

This manuscript reports on the fabrication of a hybrid surface plasmon resonance (SPR) sensor based on a gold film covered by high-density grain boundaries (GBs) monolayer (1L) WS₂ film for mercury ion (Hg²⁺) detection. The authors demonstrated the importance of the GBs as active sites for the selective and highly sensitive detection of Hg²⁺, reaching an astonishing limit of detection in the attomolar range. However there is not much novelty in this work because the CVD growth of wafer-scale polycrystalline 1L W(Mo)S₂ films has been previously published by the same authors. The main figures of merit of the present work are the selectivity and the superior sensitivity as compared to other reported substrates. Overall, I would recommend the publication of this contribution only when the following issues are addressed.

The authors reported the resonance angle shifts as a function of Hg²⁺ concentration showing that they can effectively detect Hg²⁺ in aqueous solutions at concentrations ranging from 10⁻¹¹ to 10⁻¹⁸ M (Figure 2). Could the authors comment if and how Hg²⁺ quantification is possible from these data? It appears that figure 2b reveals three different regions: i) A saturated region, where the SPR response is independent of the Hg²⁺ concentration ii) a quantification region where the response seems to be linear with the Hg²⁺ concentration and iii) a detection region where there is a response to Hg²⁺ but can not be quantified.

Reply:

We thank the reviewer for this interesting question. For quantitative analysis using a SPR metal ion sensor, we first need to determine the calibration curve of the target analyte, using a series of its standard solution. The calibration curve establishes the quantitative relationship between sensor response and analyte concentration, from which the concentration of target analyte in an unknown sample can be determined.

In our case of the GB-based SPR sensor for Hg^{2+} detection, the determined resonance angle shift ($\Delta\theta$, sensor response) as a function of Hg^{2+} concentration is the calibration curve, as shown in Fig. 2b. The most sensitive and closely linear response occurs in the concentration range from 10^{-17} to 10^{-13} M ($\Delta\theta = 26.7 \cdot \log[\text{Hg}^{2+}] + 477.5$, $R^2 = 0.9801$). For any detected Hg^{2+} concentration in this range, Hg^{2+} quantification can be determined from sensor response. For the Hg^{2+} concentration below 10^{-17} M, though the sensor response becomes weaker and exact Hg^{2+} quantification is not easy, semiquantitative analysis regarding the order of detected Hg^{2+} concentration is still achievable from the resonance angle shift $\Delta\theta$. With Hg^{2+} concentration above 10^{-13} M, the sensor response gradually tends to be saturated. Before the saturation, it is still possible to estimate the order of detected Hg^{2+} concentration. When the detected Hg^{2+} concentration becomes much higher than 10^{-11} M, however, even identification of the order will become quite difficult, and it can only be determined if the detected Hg^{2+} concentration is higher than 10^{-11} M or not. The corresponding discussion is added to the main text.

The authors have shown that their system is selective towards Hg^{2+} in the presence of common interfering ions such as Pb^{2+} , Mn^{2+} , Cu^{2+} , Fe^{3+} , Zn^{2+} , Co^{2+} , Cr^{3+} and Mg^{2+} . However, the selectivity of the presented sensor is based on the Hg-S bond formation and hence any other species than can interact with sulfur will also respond (e.g. Ag^+ or As^{3+}). The authors should clarify that in the manuscript.

Reply:

We thank the reviewer for this constructive suggestion. In response to the comment, additional investigations have been performed: 1) on sensor response to individual ion solutions (Ag^+) with possible affinity for sulphur, and 2) on the sensor response to the mixed solutions of Hg^{2+}

and Ag^+ ions for examination of any possible interference from the coexisting Ag^+ ions.

Sensor response ($\Delta\theta$) to 10^{-12} M Ag^+ is about one third of that of 10^{-12} M Hg^{2+} , yet non-negligible. Therefore, further interference studies were conducted. A series of mixed solutions of coexisting Hg^{2+} and Ag^+ ions have been prepared, in which Hg^{2+} concentration is fixed at 10^{-15} M and Ag^+ concentration is changed from 10^{-16} to 10^{-14} M. The examinations have indicated the negligible interference from the coexisting Ag^+ ions to the sensor response, as the change in $\Delta\theta$ with increasing Ag^+ from 10^{-16} M to 10^{-14} M, is only 0.56 % (Fig. 4c), which can be attributed to the preferential or selective adsorption of Hg^{2+} . These results have been added in the revised manuscript. As to As^{3+} ion, we are not able to purchase any As^{3+} -related chemicals due to its high toxicity and strict control from government.

Fig. 4. Selectivity of Hg^{2+} detection in mixed solutions of coexisting Hg^{2+} and Pb^{2+} ions **(b)** and of coexisting Hg^{2+} and Ag^+ ions **(c)**. In the mixed solutions, Hg^{2+} concentration is fixed at 10^{-15} M, and Pb^{2+} (Ag^+) is changed from 10^{-16} to 10^{-14} M. $\Delta\theta$ is the resonance angle shift.

Could the authors comment on the response time of their sensors? Is the signal collected after the immediate exposure of the substrate to the Hg²⁺ solution or is there any delay time? If so, the authors should indicate it.

Reply:

We thank the reviewer for this interesting question. The sensor response to the detected solutions has been monitored via Kinetic curve before any signal collection. Similar to any other SPR ion sensor, the sensor response generally requires a delay or response time of several minutes to reach the dynamical equilibrium for any signal collection. Therefore, there exists a response time in Hg²⁺ detection of the GB-based SPR sensor, which is ~ 2 min. The signal was collected after the response time. In the revised manuscript, Kinetic curve for the sensor response to 10⁻¹⁵ M Hg²⁺ solution has been added in Supplementary Fig. 15 to clarify the sensor response time.

Supplementary Fig. 15. Response of one GB-based SPR sensor to 10⁻¹⁵ M Hg²⁺ solution monitored by the Kinetic curve for more than 2 hours. Inset is the

amplified Kinetic curve to show the initial response. The arrow-pointed jump is induced by injection of Hg^{2+} solution.

The assessment of a real matrix (e.g. tap water) for Hg^{2+} sensing would be of high interest to demonstrate that the robustness of sensor's performance.

Reply:

We thank the reviewer for this constructive suggestion. We agree with the high significance of considering a real case, for example, the Hg^{2+} detection in tap water. However, tap water involves the more complex pollutants, including not only the possible heavy metal ions but many other possible types of impurities such as organic impurities, germs, viruses, etc. For the practical application of the GB-based SPR sensor in Hg^{2+} detection of tap water, a lot of examinations are required to be done on the possible interference to the sensor response from the more complex types of pollutants, which is hard to be completely achieved by the existing instruments and equipment in our lab. Some cooperation with the other researchers is required. Unfortunately, due to the recently increasing COVID-19 cases in China, winter vacation has to be started far ahead of schedule at most of the Universities, and it will last more than two months, much longer than the scheduled one. This makes it difficult to schedule the cooperation before the required deadline of resubmission for completion of these examinations.

From a fundamental research perspective, it is of significance to find that the GB-based SPR sensor is able to reach a 1 aM-level sensitivity in Hg^{2+} detection with high selectivity, because it demonstrates the great potential of GBs of 2D materials in fabrication of ultra-sensitive ion sensors. The GBs-rich WS_2 provides enough active sites to adsorb Hg^{2+} ions and the ultrathin thickness of 2D materials makes them very

sensitive to the adsorbed species, which will be complementary to the sensors based on traditional materials.

Definitely, we will study the possible interference to the sensor response from the more complex types of pollutants such as organic impurities, germs, viruses, etc. in the future. With systematic studies on these issues, we will further promote the practical application of GBs-rich WS₂ based SPR sensor. The corresponding discussion is added to the main text.

Minor issues:

Figure S4 is cited before S3 in the manuscript. The numbering should be reversed.

Reply:

We thank the reviewer for pointing out this issue. This has been corrected in the revised manuscript.

Did the authors test the response of their sensor with higher concentrations (>10⁻¹¹ M) of Hg²⁺? Or above this concentration the sensor's response is saturated?

Reply:

We thank the reviewer for this question. We had tested the sensor response in the Hg²⁺ solution of the higher concentration than 10⁻¹¹ M, and it was observed to reach saturation. This has been clarified in supplementary information of the revised manuscript.

Did the authors test the recyclability of their substrates?

Reply:

We thank the reviewer for this interesting question. We have tested the recyclability of sensor. In consideration of the Hg²⁺ adsorption via S-Hg bond formation on GBs, the GB-rich polycrystalline 1L WS₂ film itself, cannot be reused as sensing material after detection. The Au film substrate, however, can definitely be reused. Since the weak attachment of 1L WS₂ film on Au substrate via weak van der Waals forces, the recyclability of Au film can be realized just by ultrasonic removal of 1L WS₂ film in water. In the revised manuscript, the reusability of Au film as substrate has been clarified in supplementary information.

REVIEWER COMMENTS

Reviewer #1 (Remarks to the Author):

Further comments:

1. The GBs in WS2 and the wide size distributions of WS2 still bring device-to-device variations, which I think will greatly prevent the practical use of this type of sensor device.
2. The selectivity is still a big concern. The Ag⁺ and Pb²⁺ signals are too strong and are not easy to be differentiated from the Hg₂⁺ signal. Moreover, in real samples, the concentration of other common ions (e.g., Fe³⁺, Mg²⁺, Ca²⁺) are several orders higher than that of Hg₂⁺ (10⁻¹⁵ M), therefore, their impact will be significant. Authors tested 10⁻⁶ M Fe³⁺, Mg²⁺, Ca²⁺, Zn²⁺, and their responses are one fifth of the Hg₂⁺. However, the selectivity coefficient (signal ratio) should be at least 10 for a selective sensor and better higher than 20. Therefore, the sensor selectivity is not good.
3. Authors discussed that "the reusability of Au film as substrate can be realized just by ultrasonic removal of 1L WS2 film", but this is not a practical method since the "ultrasonic removal" treatment is not controllable. I guess this is more like a one-time use sensor, and this brings another issue of high cost.

Reviewer #2 (Remarks to the Author):

The authors replied satisfactorily to the reviewers' questions and they added all the necessary new information. The paper can be published.

Only one comment:

They refer that "The fresh as prepared ultrapure water has an initial pH value of ~7, but is observed to lower and stabilize at ~6.1 several hours after preparation".

As this pH-shift is probably due to atmospheric carbon dioxide absorption, probably they could try to use inert atmosphere to achieve a stable pH value near 7.

Nikolaos Kallithrakas-Kontos
Professor

Reviewer #3 (Remarks to the Author):

The authors have significantly revised the manuscript, however there are few issues that still need to be addressed:

Major issues:

Regarding Figure 2b, the authors have confirmed the existence of the three different regimes in the detection of Hg₂⁺. i) Higher concentrations than 10⁻¹³ M lead to the saturation of their sensor. ii) Within the range from 10⁻¹³ to 10⁻¹⁷ M, Hg₂⁺ concentration can be quantitatively expressed by the empirical formula: $\Delta\theta = 26.7 \cdot \log[\text{Hg}_2^+] + 477.5$, $R^2 = 0.9801$. iii) Below 10⁻¹⁷ M, Hg₂⁺ concentration can not be quantified but it can still be detected. I would recommend the authors to report the empirical formula in the main text. Besides, when the authors showed the Hg₂⁺ sensing in tap water (Figure S18b) it seems that up to 10⁻¹⁴ M all the curves have the same slope, and hence sensitivity. If a similar empirical formula is obtained in tap water, that would mean that the author's sensor sensitivity is not compromised by the real matrix.

The most critical point regarding selectivity has been partially solved. The authors have

demonstrated that their sensor is highly selective in the presence of common interfering cations such as Mn^{2+} , Cu^{2+} , Ca^{2+} , Fe^{3+} , Zn^{2+} , Co^{2+} , Cr^{3+} and Mg^{2+} (Figure 4 and S12). The authors have explained the selectivity in terms of the strength of Lewis acids. The authors have conducted selectivity experiments for some of the hard acids at a concentration of 10^{-6} M due to their high abundance in water (Figure S12). The most interfering cations, Ag^{+} or Pb^{2+} , were only tested at a concentration 10X higher than Hg^{2+} showing a negligible response ($\sim 1\%$) (Figure 4b-c). However, Figure 4a where the concentration of both, Hg^{2+} and interfering cations is 10^{-12} M shows that the contribution of the interfering cations is not as negligible, being $1/3$ of the Hg^{2+} response. Hence if the concentration of Ag^{+} or Pb^{2+} were higher than the one of Hg^{2+} , it could lead to a potential false positive. The authors should explain this or complete the selectivity experiment of Figures 4b-c for higher concentrations of the interfering cations.

Minor issues:

The authors have used tap water without additional purification. Usually a filtration or centrifugation step can be very helpful to get rid of most of the impurities.

Firstly, we would like to appreciate the Reviewers' comments, based on which our revised manuscript has been significantly improved. The revisions based on the comments of the reviewers are highlighted in blue in the manuscripts.

REVIEWER COMMENTS AND OUR CORRESPONDING REPLIES

Reviewer #1 (Remarks to the Author):

Further comments:

1. The GBs in WS₂ and the wide size distributions of WS₂ still bring device-to-device variations, which I think will greatly prevent the practical use of this type of sensor device.

Reply:

We understand the reviewer's concern about the grain size distribution in the polycrystalline monolayer WS₂ film. During the investigations on the CVD growth of monolayer WS₂ film, we have found that the grain sizes and their distribution are strongly dependent on the choice of S source. When the S powder is used as the source, our studies indicate that the grains can be grown up to several hundreds of μm in size and in a wide size distribution, as shown in Supplementary Fig. 3. For the SPR sensors fabricated with the S-powder-grown 1L WS₂ film as shown in Supplementary Fig. 3, though the spot size of detection light is as large as 170 μm in diameter, Supplementary Fig. 4 shows that the large grains and the wide size distribution lead to the obvious device-to-device differences between their sensing responses to Hg²⁺ detection. We have provided these results in Supplementary Figs. 3 and 4 to demonstrate that the nanoscale grain sizes and narrow size distribution are the

prerequisites for the consistent device-to-device performances of our GB-based sensors. In the submitted work, we have taken the H₂S gas as the S source in CVD growth of 1L WS₂ films on SiO₂ (300 nm)/Si substrates, and our measurements as shown in Fig.1 have indicated that the grains can be well controlled to grow in the nanoscale average size of ~40 nm and in a narrow size distribution from 20 to 55 nm. Within the large detection spot of light (170 μm in diameter), the nanoscale 1L WS₂ grains are present in a huge amount, at least in the order of 10⁶, and therefore, the high-density grain boundaries (GBs) can be considered to be uniformly distributed over the large detection area. For the SPR sensors based on the high-density GBs, Supplementary Figs. 2 and 13, have demonstrated the consistent position-to-position and device-to-device performances, which favor the practical application of our GB-based SPR sensors.

Supplementary Fig. 2. Determined resonance angle shifts of $\Delta\theta$ from 5 different areas on the CVD-grown GB-rich polycrystalline 1L WS₂ film as sensing material in the GB-based SPR sensor. The consistent values of $\Delta\theta$ indicate the desirable growth of nanoscale grains for consistent detection of Hg²⁺ from the different locations of the film because the spot size of detection light (170 μm in diameter) is much greater than nanoscale grain dimensions.

Supplementary Fig. 13. Resonance angle shifts of $\Delta\theta$, which were determined from the responses of 15 GB-based SPR sensors to the 10^{-15} M Hg^{2+} solution. These sensors were fabricated with GB-rich polycrystalline 1L WS_2 films from 15 batches grown at different dates. The consistency in values of $\Delta\theta$ and small standard deviation indicates the high repeatability ($\Delta\theta = 89.28 \pm 2.89$ mdeg).

2. The selectivity is still a big concern. The Ag^+ and Pb^{2+} signals are too strong and are not easy to be differentiated from the Hg^{2+} signal. Moreover, in real samples, the concentration of other common ions (e.g., Fe^{3+} , Mg^{2+} , Ca^{2+}) are several orders higher than that of Hg^{2+} (10^{-15} M), therefore, their impact will be significant. Authors tested 10^{-6} M Fe^{3+} , Mg^{2+} , Ca^{2+} , Zn^{2+} , and their responses are one fifth of the Hg^{2+} . However, the selectivity coefficient (signal ratio) should be at least 10 for a selective sensor and better higher than 20. Therefore, the sensor selectivity is not good.

Reply:

Thanks to the comments of Reviewer 1 and 3, we wake up to find that there was a bug in the previous evaluation of sensor selectivity by using the higher Hg^{2+} concentration of 10^{-12} M as shown in original Figure 4a. As revealed in Figure 2b, the exact quantitative detection of Hg^{2+} ions can be achieved in the Hg^{2+} concentration range from 10^{-17} to 10^{-13} M. Yet, above 10^{-13} M, quantitative detection of Hg^{2+} is not possible since sensor

response tends to saturate. Thus, the large uncertainty would be created in the previous evaluation of sensor selectivity by using the 10^{-12} M Hg^{2+} solution as reference.

In the revised manuscript, a series of S0-S7 solutions have been re-prepared for the more detailed and comprehensive investigations on the sensor selectivity. S0 represents the 10^{-15} M Hg^{2+} solution as reference. S1(S2) are the solutions of 10^{-15} M Hg^{2+} and $\text{Pb}^{2+}(\text{Ag}^+)$, in which the $\text{Pb}^{2+}(\text{Ag}^+)$ concentrations are varied from 10^{-15} M to 10^{-12} M. S3-S6 are the solutions of 10^{-15} M Hg^{2+} and one of the naturally abundant interfering Zn^{2+} , Fe^{3+} , Ca^{2+} and Mg^{2+} ions, in which all the interfering ion concentrations are varied from 10^{-8} M to 10^{-5} M. S7 is the solution of 10^{-15} M Hg^{2+} with mixed interfering ions, including Pb^{2+} , Ag^+ at 10^{-12} M and Zn^{2+} , Fe^{3+} , Ca^{2+} , Mg^{2+} at 10^{-5} M. The sensor responses to the re-prepared S0-S7 solutions have been examined, and the extracted resonance angle shifts of $\Delta\theta$ are used as the revised Fig.4 to replace the previous one.

Compared with that of S0 (10^{-15} M Hg^{2+}) as reference, the sensor responses to the S1(S2) solutions exhibit just a slight increase in resonance angle shift $\Delta\theta$ when the $\text{Pb}^{2+}(\text{Ag}^+)$ concentration is increased from 10^{-15} M to 10^{-12} M, implying the negligible dependence of $\Delta\theta$ on $\text{Pb}^{2+}(\text{Ag}^+)$ concentration and thus the negligible interference of $\text{Pb}^{2+}(\text{Ag}^+)$ ions to Hg^{2+} detection of the GB-based sensors. This is further corroborated with XPS analysis of sensor material after detection in mixed Hg^{2+} (10^{-15} M) and Pb^{2+} (10^{-12} M) solutions. The XPS spectra exhibit the appearance of a distinct Hg 4f core level peak around the binding energy of 102 eV but no observable sign of Pb 4f core level peak around the binding energies of ~ 138 and ~ 143 eV (Supplementary Fig. 12). Thereby, the interference to Hg^{2+} detection from the coexisting $\text{Pb}^{2+}(\text{Ag}^+)$ ions could be neglected even at a 1000X higher concentration of interfering ions. Additionally, compared with S0 (10^{-15} M Hg^{2+}) as reference, the S3-S6 solutions lead to no significant variations of $\Delta\theta$

when the concentrations of Zn^{2+} , Fe^{3+} , Ca^{2+} or Mg^{2+} ions are increased from 10^{-8} M to 10^{-5} M. Even in the presence of mixed interfering ions (Pb^{2+} , Ag^+ , Zn^{2+} , Fe^{3+} , Ca^{2+} and Mg^{2+}), the S7 solution does not produce an obvious change in the sensor response in comparison to the reference solution S0, and just a negligible increase in $\Delta\theta$ is observable. These detailed results indicate the satisfactory selectivity for Hg^{2+} detection of the GB-based sensor.

Fig. 4. Interference study of Hg^{2+} detection. GB-based SPR sensor response ($\Delta\theta$) to Hg^{2+} at a fixed concentration (10^{-15} M) with increasing concentration of interfering ions (Pb^{2+} , Ag^+ , Zn^{2+} , Fe^{3+} , Ca^{2+} and Mg^{2+}). S0 is the 10^{-15} M Hg^{2+} solution as reference. S1(S2) are the solutions of 10^{-15} M Hg^{2+} and Pb^{2+} (Ag^+), the Pb^{2+} (Ag^+) concentration is varied from 10^{-15} M to 10^{-12} M. S3-S6 are the solutions of 10^{-15} M Hg^{2+} and one of the naturally abundant interfering Zn^{2+} , Fe^{3+} , Ca^{2+} and Mg^{2+} ions, the Zn^{2+} , Fe^{3+} , Ca^{2+} , Mg^{2+} concentration is varied from 10^{-8} M to 10^{-5} M. S7 is the solution of 10^{-15} M Hg^{2+} with mixed interfering ions, including Pb^{2+} , Ag^+ at 10^{-12} M and Zn^{2+} , Fe^{3+} , Ca^{2+} , Mg^{2+} at 10^{-5} M.

3. Authors discussed that “the reusability of Au film as substrate can be realized just by ultrasonic removal of 1L WS2 film”, but this is not a practical

method since the “ultrasonic removal” treatment is not controllable. I guess this is more like a one-time use sensor, and this brings another issue of high cost.

Reply:

We understand the reviewer’s concern about the high cost. In our studies, several pieces of Au films have been first prepared by magnetron sputtering, and the as-prepared Au films have been used in the first round of Hg²⁺ detection measurements. After the first round of use, the Au films have been recycled for the more subsequent rounds of detection. Actually, we have been using the recycled Au films to prepare the required additional data in the second and third revised manuscripts. In our submitted work, before the preparation of any GB-based sensor, even the as-prepared Au films were ultrasonically cleaned in order to remove any possible absorbed contaminants on the surface due to exposure to air. In our investigations, we have recycled the Au films after Hg²⁺ detection measurements via the ultrasonic removal of the 1L WS₂ film, and simultaneously, we have gotten the film surface ultrasonically cleaned. For comparison, Fig. R1 shows the optical images for one piece of as-grown Au film and one piece of recycled Au film after the ninth round of Hg²⁺ detection with the transferred 1L WS₂ film on top and after removal. Even after the ninth round of detection and ultrasonic removal, the Au film shows no observable damage, and it can be further reused, indicating the recyclability.

Fig. R1. Photograph of Au film substrates (left to right): (a) new; (c) 9X reused with the 1L WS₂ film on top; (e) cleaned after 9X reuse. The corresponding optical microscopy images are shown in (b), (d) and (f), respectively.

Reviewer #2 (Remarks to the Author):

The authors replied satisfactorily to the reviewers' questions and they added all the necessary new information. The paper can be published.

Only one comment:

They refer that "The fresh as prepared ultrapure water has an initial pH value of ~7, but is observed to lower and stabilize at ~6.1 several hours after preparation".

As this pH-shift is probably due to atmospheric carbon dioxide absorption, probably they could try to use inert atmosphere to achieve a stable pH value near 7.

Nikolaos Kallithrakas-Kontos

Professor

Reply:

We thank the reviewer for this suggestion, which is adopted to achieve stable pH around 7 for ultrapure water. SPR spectra of the GB-based SPR sensor (Fig. R2) verifies that pH does not impact sensor response at pH above 5.

Fig. R2. SPR spectra of the GB-based SPR sensor at different pH. pH = 7.0 is ultrapure water under N₂, pH = 6.1 is ultrapure water in air, and pH = 5.5, 5.0 are acidified water.

Reviewer #3 (Remarks to the Author):

The authors have significantly revised the manuscript, however there are few issues that still need to be addressed:

Minor issues:

The authors have used tap water without additional purification. Usually a filtration or centrifugation step can be very helpful to get rid of most of the impurities.

Reply:

We appreciate the reviewer's suggestion very much. Based on this suggestion, we have performed a simple filtration of the tap water by using filter paper, and then we have re-done the Hg^{2+} detection of GB-based sensor in the filtered tap water matrix. Just as pointed out by the Reviewer, the simple filtration is very helpful to get rid of most of the impurities, and most probably, the insoluble impurities are removed by the simple filtration. After the simple filtration, the background difference between ultrapure water and tap water is reduced (see Supplementary Fig. 17). In comparison to those in tap water without any additional purification, the sensor responses in the filtered tap water matrix have been significantly improved to give the more consistent device-to-device sensing performances, and in the revised manuscript, the corresponding obtained resonance angle shifts of $\Delta\theta$ have been shown in Supplementary Fig. 17 to replace the previously obtained ones in tap water without any additional purification.

Supplementary Fig. 17. (a) SPR spectra of GB-based SPR sensors in ultrapure water and filtered tap water using filter paper for comparison. **(b)** Resonance angle shifts of $\Delta\theta$ extracted from the SPR spectra of 3 GB-based sensors at Hg^{2+} concentrations ranging from 10^{-18} M to 10^{-11} M in the filtered tap water matrix. The red curve is the fit of the average values of three GB-based sensors.

Major issues:

Regarding Figure 2b, the authors have confirmed the existence of the three different regimes in the detection of Hg^{2+} . i) Higher concentrations than 10^{-13} M lead to the saturation of their sensor. ii) Within the range from 10^{-13} to 10^{-17} M, Hg^{2+} concentration can be quantitatively expressed by the empirical formula: $\Delta\theta = 26.7 \cdot \log[Hg^{2+}] + 477.5$, $R^2 = 0.9801$. iii) Below 10^{-17} M, Hg^{2+} concentration can not be quantified but it can still be detected. I would recommend the authors to report the empirical formula in the main text. Besides, when the authors showed the Hg^{2+} sensing in tap water (Figure S18b) it seems that up to 10^{-14} M all the curves have the same slope, and hence sensitivity. If a similar empirical formula is obtained in tap water, that would mean that the author's sensor sensitivity is not compromised by the real matrix.

Reply:

As shown in the updated Supplementary Fig. 17, the three tested GB-sensors give robust and quite consistent responses to the prepared solutions of Hg^{2+} ions, and a clear positive correlation between Hg^{2+} concentration in tap water and sensor response can be achieved. For the three tested GB-sensors, their average value of $\Delta\theta$ can be expressed by an empirical formula of $\Delta\theta = 19.2 \cdot \log[\text{Hg}^{2+}] + 356.7$ ($R^2 = 0.9820$) within the Hg^{2+} concentration range of 10^{-17} M to 10^{-13} M. In comparison to $\Delta\theta = 26.7 \cdot \log[\text{Hg}^{2+}] + 477.5$ ($R^2 = 0.9801$) obtained in ultrapure water, though the slope of 19.2 in the filtered tap water matrix becomes smaller, it still keeps the same order, and more importantly, as revealed in the updated Supplementary Fig.17, the obvious sensor response ($\Delta\theta$) at 1 aM is still achieved in the real tap water matrix. Thereby, the sensitivity of GB-based sensor is not compromised very much in the real tap water matrix after a simple filtration with filter paper, and the robust performance and applicability of the GB-based sensor can be guaranteed in real tap water. As recommended by the Reviewer, we have added the obtained the empirical formula for both ultrapure water and filtered tap water matrix and the corresponding discussions in the revised manuscript.

The most critical point regarding selectivity has been partially solved. The authors have demonstrated that their sensor is highly selective in the presence of common interfering cations such as Mn^{2+} , Cu^{2+} , Ca^{2+} , Fe^{3+} , Zn^{2+} , Co^{2+} , Cr^{3+} and Mg^{2+} (Figure 4 and S12). The authors have explained the selectivity in terms of the strength of Lewis acids. The authors have conducted selectivity experiments for some of the hard acids at a concentration of 10^{-6} M due to their high abundance in water (Figure S12). The most interfering cations, Ag^+ or Pb^{2+} , were only tested at a concentration 10X higher than Hg^{2+} showing a negligible response ($\sim 1\%$) (Figure 4b-c). However, Figure 4a where the

concentration of both, Hg^{2+} and interfering cations is 10^{-12} M shows that the contribution of the interfering cations is not as negligible, being 1/3 of the Hg^{2+} response. Hence if the concentration of Ag^+ or Pb^{2+} were higher than the one of Hg^{2+} , it could lead to a potential false positive. The authors should explain this or complete the selectivity experiment of Figures 4b-c for higher concentrations of the interfering cations.

Reply:

Thanks to the comments of Reviewer 3 and 1, we wake up to find that there was a bug in the previous evaluation of sensor selectivity by using the higher Hg^{2+} concentration of 10^{-12} M as shown in original Figure 4a. As revealed in Figure 2b, the exact quantitative detection of Hg^{2+} ions can be achieved in the Hg^{2+} concentration range from 10^{-17} to 10^{-13} M. Yet, above 10^{-13} M, quantitative detection of Hg^{2+} is not possible since sensor response tends to saturate. Thus, the large uncertainty would be created in the previous evaluation of sensor selectivity by using the 10^{-12} M Hg^{2+} solution as reference.

In the revised manuscript, a series of S0-S7 solutions have been re-prepared for the more detailed and comprehensive investigations on the sensor selectivity. S0 represents the 10^{-15} M Hg^{2+} solution as reference. S1(S2) are the solutions of 10^{-15} M Hg^{2+} and Pb^{2+} (Ag^+), in which the Pb^{2+} (Ag^+) concentrations are varied from 10^{-15} M to 10^{-12} M. S3-S6 are the solutions of 10^{-15} M Hg^{2+} and one of the naturally abundant interfering Zn^{2+} , Fe^{3+} , Ca^{2+} and Mg^{2+} ions, in which all the interfering ion concentrations are varied from 10^{-8} M to 10^{-5} M. S7 is the solution of 10^{-15} M Hg^{2+} with mixed interfering ions, including Pb^{2+} , Ag^+ at 10^{-12} M and Zn^{2+} , Fe^{3+} , Ca^{2+} , Mg^{2+} at 10^{-5} M. The sensor responses to the re-prepared S0-S7 solutions have been examined, and the extracted resonance angle shifts of $\Delta\theta$ are used as the revised Fig. 4 to replace the previous one.

Compared with that of S0 (10^{-15} M Hg^{2+}) as reference, the sensor responses to the S1(S2) solutions exhibit just a slight increase in

resonance angle shift $\Delta\theta$ when the $\text{Pb}^{2+}(\text{Ag}^+)$ concentration is increased from 10^{-15} M to 10^{-12} M, implying the negligible dependence of $\Delta\theta$ on $\text{Pb}^{2+}(\text{Ag}^+)$ concentration and thus the negligible interference of $\text{Pb}^{2+}(\text{Ag}^+)$ ions to Hg^{2+} detection of the GB-based sensors. This is further corroborated with XPS analysis of sensor material after detection in mixed Hg^{2+} (10^{-15} M) and Pb^{2+} (10^{-12} M) solutions. The XPS spectra exhibit the appearance of a distinct Hg 4f core level peak around the binding energy of 102 eV but no observable sign of Pb 4f core level peak around the binding energies of ~ 138 and ~ 143 eV (Supplementary Fig. 12). Thereby, the interference to Hg^{2+} detection from the coexisting Pb^{2+} (Ag^+) ions could be neglected even at a 1000X higher concentration of interfering ions. Additionally, compared with S0 (10^{-15} M Hg^{2+}) as reference, the S3-S6 solutions lead to no significant variations of $\Delta\theta$ when the concentrations of Zn^{2+} , Fe^{3+} , Ca^{2+} or Mg^{2+} ions are increased from 10^{-8} M to 10^{-5} M. Even in the presence of mixed interfering ions (Pb^{2+} , Ag^+ , Zn^{2+} , Fe^{3+} , Ca^{2+} and Mg^{2+}), the S7 solution does not produce an obvious change in the sensor response in comparison to the reference solution S0, and just a negligible increase in $\Delta\theta$ is observable. These detailed results indicate the satisfactory selectivity for Hg^{2+} detection of the GB-based sensor.

Fig. 4. Interference study of Hg²⁺ detection. GB-based SPR sensor response ($\Delta\theta$) to Hg²⁺ at a fixed concentration (10^{-15} M) with increasing concentration of interfering ions (Pb²⁺, Ag⁺, Zn²⁺, Fe³⁺, Ca²⁺ and Mg²⁺). S0 is the 10^{-15} M Hg²⁺ solution as reference. S1(S2) are the solutions of 10^{-15} M Hg²⁺ and Pb²⁺(Ag⁺), the Pb²⁺(Ag⁺) concentration is varied from 10^{-15} M to 10^{-12} M. S3-S6 are the solutions of 10^{-15} M Hg²⁺ and one of the naturally abundant interfering Zn²⁺, Fe³⁺, Ca²⁺ and Mg²⁺ ions, the Zn²⁺, Fe³⁺, Ca²⁺, Mg²⁺ concentration is varied from 10^{-8} M to 10^{-5} M. S7 is the solution of 10^{-15} M Hg²⁺ with mixed interfering ions, including Pb²⁺, Ag⁺ at 10^{-12} M and Zn²⁺, Fe³⁺, Ca²⁺, Mg²⁺ at 10^{-5} M.

REVIEWERS' COMMENTS

Reviewer #1 (Remarks to the Author):

The authors have supplemented with new data. The manuscript can be accepted for publication.

Reviewer #3 (Remarks to the Author):

At this stage I feel that the manuscript can be accepted for publication. The authors tackled all the comments raised by me and other referees, and the quality of the manuscript improved significantly since its first version. I have no further comments.